# NIMA-related kinase 9 regulates the phosphorylation of the essential myosin light chain in the heart

Marion Müller [1,2,3], Rose Eghbalian[1,2], Jes-Niels Boeckel [4], Karen S. Frese [2,5], Jan Haas[2,5], Elham Kayvanpour[2,5], Farbod Sedaghat-Hamedani [2,5], Maximilian K. Lackner[2,5], Oguz F. Tugrul[2,5], Thomas Ruppert[6,7], Rewati Tappu[2,5], Diana Martins Bordalo[2,5], Jasmin M. Kneuer [4], Annika Piekarek [5], Sabine Herch[5], Sarah Schudy[2,5], Andreas Keller [8,9], Nadja Grammes[8,9], Cornelius Bischof [3], Anna Klinke[3], Margarida Cardoso-Moreira [7], Henrik Kaessmann [7], Hugo A. Katus[2,5], Norbert Frey[2,5], Lars M. Steinmetz [2,10] & Benjamin Meder [1,2,10] ✉

To adapt to changing hemodynamic demands, regulatory mechanisms modulate actin-myosin-kinetics by calcium-dependent and -independent mechanisms. We investigate the posttranslational modification of human essential myosin light chain (ELC) and identify NIMA-related kinase 9 (NEK9) to interact with ELC. NEK9 is highly expressed in the heart and the interaction with ELC is calcium-dependent. Silencing of NEK9 results in blunting of calcium-dependent ELC-phosphorylation. CRISPR/Cas9-mediated disruption of NEK9 leads to cardiomyopathy in zebrafish. Binding to ELC is mediated via the protein kinase domain of NEK9. A causal relationship between NEK9 activity and ELC-phosphorylation is demonstrated by genetic sensitizing invivo. Finally, we observe significantly upregulated ELC-phosphorylation in dilated cardiomyopathy patients and provide a unique map of human ELC-phosphorylation-sites. In summary, NEK9-mediated ELC-phosphorylation is a calcium-dependent regulatory system mediating cardiac contraction and inotropy.

Cardiomyopathies are frequent genetic disorders involving more than 40 different disease genes. In the last years, much attention was paid to understand the mechanisms involved in sarcomeric dysfunction and first compounds have proven clinically effective in modulating myosin kinetics[1]. Myosin light chains (MLCs) are integral parts of the sarcomere located at the myosin motor complex[2]. The MLC protein family consists of two classes of EF-hand calcium ($Ca^{2+}$) -binding proteins, the regulatory cardiac myosin light chain (RLC/cMLC2) and the essential cardiac myosin light chain (ELC/cMLC1). Each MLC exhibits two cardiac isoforms in humans: RLC is encoded by *myl7* (atrial form) and *myl2* (ventricular form) and ELC is

encoded by *myl4* (atrial form) and *myl3* (ventricular form)[3]. Mutations in genes encoding MLCs account for up to 5% of inherited cardiomyopathy cases[4], but the mechanisms which render a given genotype in a phenotype are largely unknown.

Both MLC proteins, ELC and RLC, bind to the neck region of the myosin heavy chain (MHC), being directly involved in the cross-bridge cycle. The myosin neck region acts as a lever arm, transducing force generated by the catalytic motor domain[5,6]. The MLC binding region stabilizes the neck region and contributes to the conversion of chemical energy into cardiac contraction. Furthermore, the N-terminal ELC protein can interact with actin[7–10], which positively affects force

generation in patients with congenital heart disease expressing pathological levels of atrial ELC[11].

An important role is attributed to posttranslational phosphorylation of MLC proteins. For example, $Ca^{2+}$-dependent RLC-phosphorylation by the myosin light chain kinase (MLCK)[12,13] is pivotal for the initiation of contraction in smooth muscle cells[14,15]. Cardiac MLCK function is stimulated by $Ca^{2+}$ and calmodulin[16]. Consequently, RLC-phosphorylation potentiates force and speed of cardiac contraction by higher myosin-actin binding affinity[17,18]. Noticeable, relatively small alterations of cross-bridge formation can significantly affect cardiac function. Therefore, it is crucial to precisely decipher sarcomeric regulation to ultimately translate those findings into therapeutic approaches. Although highly conserved ELC-phosphorylation sites have been annotated[19,20], only recently we were able to demonstrate the functional relevance of ELC-phosphorylation in the context of cardiac contractility. Lack of ELC or the ELC-phosphorylation site at the carboxy-terminus leads to severely impaired cardiac contractility in zebrafish[21,22]. Subsequent studies underlined the relevance of ELC-phosphorylation during physiological and pathological cardiac stress. In detail, the level of ELC-phosphorylation significantly increased during forced swimming of adult zebrafish, which is accompanied by increased sliding velocity of actin filaments on immobilized myosin proteins in-vitro[23]. Taken together, the ELC-phosphorylation was shown to be responsible not only for maintenance of cardiac force generation, but also for the modulation of sarcomere function during stress[21,23].

Here, we aim to identify potential ELC kinases and analyze ELC-phosphorylation sites in the human heart. NIMA-related kinase 9 (NEK9) interacts with ELC and is highly expressed in the heart. NEK9-mediated ELC-phosphorylation is a $Ca^{2+}$-dependent mechanism and influences cardiac function in-vivo. ELC-phosphorylation is detected in the human heart and upregulated in systolic heart failure. The ELC-phosphorylation status in patients inversely correlates with the left ventricular ejection fraction. Understanding the regulation of ELC-phosphorylation in human myocardium will help to develop targeted pharmaceutical interventions.

## Results

### NEK9 and CamK2G interact with ELC in the heart

To identify ELC interacting proteins, we covalently coupled custom-specific ELC antibodies to magnetic beads for immunoprecipitation (IP). Identification of precipitated proteins was performed by liquid chromatography followed by mass spectrometry (LC-MS) analysis (Fig. 1A–D). In total, 110 proteins were identified as potential interaction partners. In brief, amongst the 106 ELC interacting proteins, 33% are annotated as enzymes and 34% are assigned to structural proteins (Fig. 1B). Of structural proteins, 64% are sarcomere-associated proteins and known ELC interactors, such as myosin binding protein C (MyBPC3), cardiac myosin heavy chain 6 (MYH6), troponin I type 1b (TNNI1b) and ventricular myosin heavy chain (MYH7) (Fig. 1C). Of the annotated enzymes, 3 kinases were identified. NIMA-related kinase 9 (NEK9) and calcium/calmodulin-dependent protein kinase type II subunit gamma (CamK2G) are both serine/threonine kinases with high cross-species homology (Supplementary Fig. 1A, B). Phospho-fructokinase was identified by only two distinct peptides and was not further analyzed. By contrast, NEK9 and CamK2G could be identified as ELC interacting kinases with multiple hits and different unique peptides by LC-MS (Fig. 1D and Supplementary Fig. 1). No predicted or known phosphatase was captured by MS screening. The interaction of NEK9, as well as CamK2G with ELC, could be validated in human cell cultures using kinase specific IP followed by immunoblotting (IB). Both kinases were found to specifically interact with ELC (Fig. 1E). NEK9 was precipitated at much higher amounts considering the almost equal expression levels of NEK9 and CamK2G (Fig. 1E: input). Further, the approach was confirmed by an independent interaction assay using

ascorbate peroxidase (APEX) catalyzed proximity labeling, showing NEK9 and CamK2G to interact with ELC (Fig. 1F). Together, these experiments were able to identify an ELC interacting kinase with yet unknown cardiac function.

### NEK9 is enriched in the left ventricle

Next, we characterized the expression of NEK9 and CamK2G in a human tissue RNA panel (Fig. 2A) and by specific IB on the protein level (Fig. 2B). NEK9 was highly expressed in human heart tissue on the mRNA (Fig. 2A) and protein level (Fig. 2B) as well as in human skeletal muscle on mRNA level (Fig. 2A). As shown previously[24,25], CamK2G expression levels are highest in brain and skeletal muscle tissue (Fig. 2A). In contrast to NEK9, CamK2G was found to be less abundantly expressed in the human heart on the protein (Fig. 2B) and mRNA levels, but with equal expression in human left ventricular (LV) tissue and atrium. NEK9 was found to be highly enriched in human LV tissue compared to the atrium (Fig. 2A). These data could be confirmed by developmental mRNA expression analysis, showing strong expression of NEK9 in the heart from 4 weeks post-conception to adulthood, while cardiac CamK2G expression remains at a low expression level throughout human development (Fig. 2C). To demonstrate cross-species applicability of our results, NEK9 protein expression was also investigated in larval and adult zebrafish heart (Fig. 2B; left panel). Taken together, in contrast to CamK2G, NEK9 expression is enriched in human LV tissue and during adulthood, rendering NEK9 an interesting candidate for further functional investigations.

### The interaction of ELC with its kinases is dependent on calcium

To further evaluate the mode of interaction of ELC with the identified NEK9 kinase candidate, plasmid overexpression of human myc-tagged ELC protein was performed in human cells followed by myc-specific IP and kinase specific IB. Importantly, ELC is not expressed in HEK293 cells (Fig. 2D, first line showing untransfected cells), which increases specificity. NEK9 interacts with ELC (Fig. 2D, E) under basal conditions, underlining the reproducibility of our ELC interaction screening (Fig. 1). As calcium ($Ca^{2+}$)-influx is mandatory for cardiac muscle contraction, we analyzed whether increasing $Ca^{2+}$ concentrations influence the kinase-target interaction. Figure 2D, E presents a 3-fold increase of interacting NEK9 protein by doubling the ionophore concentration ($0.35 \pm 0.18$ vs. $1.01 \pm 0.13$). As demonstrated, NEK9/ELC interaction is $Ca^{2+}$-sensitive, potentially enabling increased interaction of ELC and its kinase during physical exercise or under pathological conditions that go along with high intracellular $Ca^{2+}$-load.

### NEK9 mediates ELC-phosphorylation

To analyze the impact of NEK9 on ELC-phosphorylation, siRNA-mediated knockdown of NEK9 was conducted in human cells after myc-tagged ELC overexpression. Knockdown of NEK9 led to a significant reduction of NEK9 kinase mRNA (Fig. 3A) and protein levels (Fig. 3B). As expected, the efficiency of the knockdown was independent from the stimulation with ionomycin and $Ca^{2+}$-influx (Fig. 3A, B). Next, the impact on ELC-phosphorylation was detected by 2D immunoblot against the myc-tagged human ELC form. In brief, isolated protein lysate was separated by isoelectric point (pI) and molecular weight (MW). Phosphorylated ELC proteins contain additional negatively charged phosphates, leading to a shift towards an acidic pI (see also Fig. 7). Besides the most abundant basic ELC spot, two additional spots shifted towards the acidic pI can be detected by IB (Fig. 3C, D). For validation, we used myc-specific and ELC-specific antibodies for one sample in a reblotting strategy, which resulted in the same ELC-phosphorylation pattern (Supplementary Fig. 2). Untransfected HEK293 cells did not show any ELC nor myc-tag signal, while equal amounts of the standard protein β-actin were detected in the transfected compared to the untransfected sample (Supplementary Fig. 2). Importantly, the myc-tagged human ELC resembled the ELC-

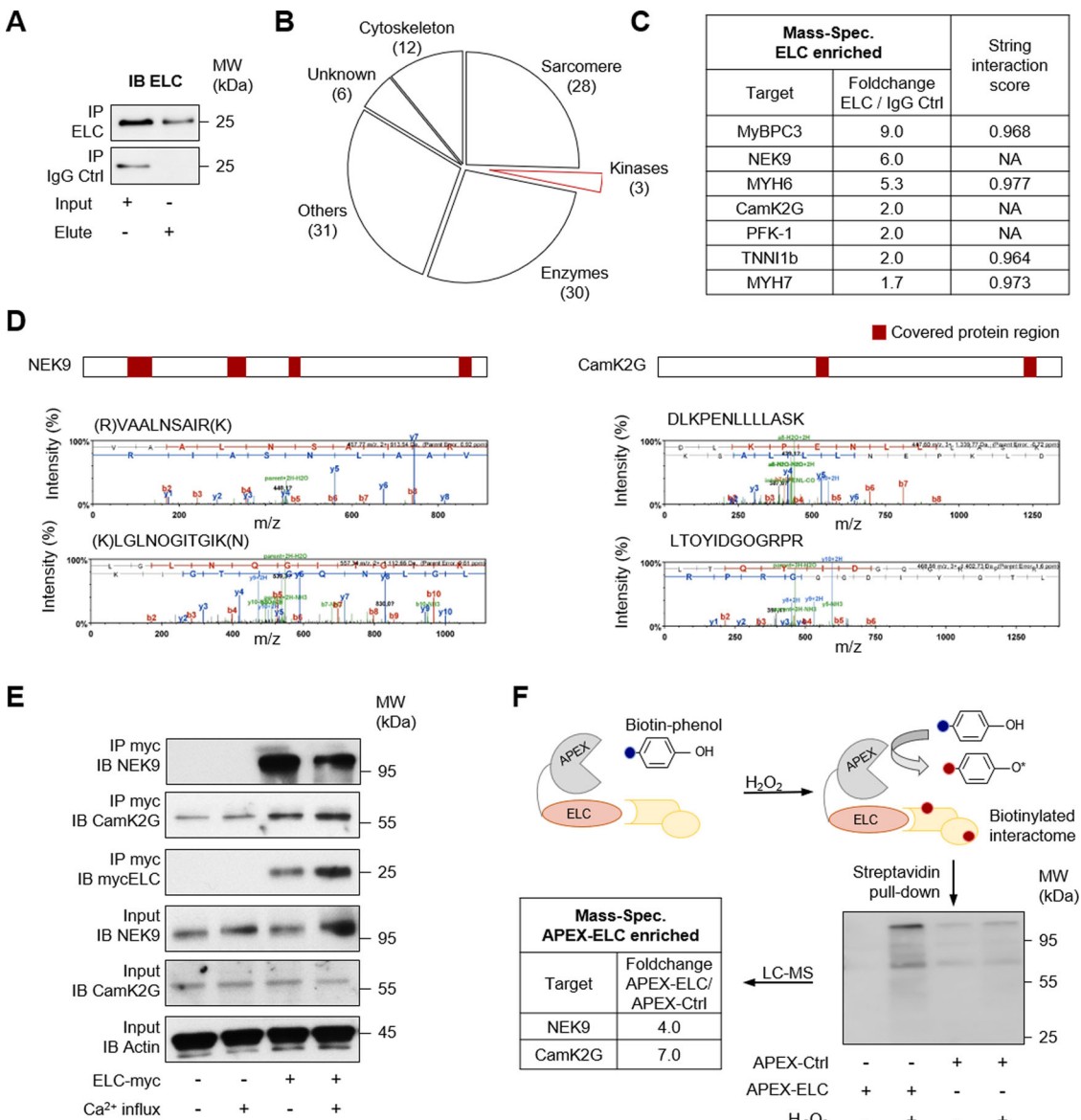

**Fig. 1 | Identification of ELC kinases in adult piscine heart protein.** Immuno-precipitation (IP) was performed by using custom ELC antibodies covalently coupled to magnetic beads. Liquid chromatography followed by mass spectrometry (LC-MS) was used for identification. **A** Validation of ELC precipitation by immunoblot (IB) before performing LC-MS (see **B** and **C**). **B** Classification of ELC enriched proteins identified by LC-MS. From 3452 spectra 110 proteins were identified in total. 106 proteins were found to interact with ELC. 3 different kinases were found. (Peptide FDR: 5%, minimum of two identified peptides per protein) **C** Ranking of ELC enriched proteins by foldchange of unique peptide counts enriched by ELC compared to unique peptide counts of the same protein captured by the isotype control (IgG Ctrl). Data of the screening experiment are shown without statistical analysis. Candidates were validated using independent methods. The string intersection score was calculated for known ELC interacting proteins[67]. (NA: not available) **D** Two representative mass spectrometry spectra of unique peptides accepted to identify NIMA related kinase 9 (NEK9) (left panel) or calcium/calmodulin-dependent protein kinase type II subunit gamma (CamK2G) (right panel). The protein coverage is shown above (red boxes). **E** Validation of ELC-kinase interaction. Human myc-tagged ELC protein was overexpressed in human cells followed by specific immunoprecipitation (IP) and immunoblot (IB) analysis. $Ca^{2+}$-influx was obtained by ionophore stimulation. One representative experiment is shown (see also Fig. 2D). **F** Schematic illustration of ascorbate peroxidase (APEX) catalyzed proximity labeling. Overexpression of ELC fused to APEX enables biotin-labeling of proteins in close proximity. The reaction is catalyzed by hydrogen peroxide ($H_2O_2$). Streptavidin precipitation followed by LC-MS analysis identified NEK9 and CamK2G. The foldchange of unique peptide counts of biotin-labeled proteins in the proximity of ELC (APEX-ELC) compared to unique peptide counts of the same protein in the APEX-Ctrl sample is shown. Data of the validation experiment are shown without statistical analysis.

phosphorylation pattern in human myocardium (see also Figs. 7, 8). Phosphatase treatment resulted in loss of predicted phosphorylated forms, again underlining the specificity to detect ELC-phosphorylation (Fig. 3C). Taken together, the established ELC-phosphorylation assay is able to capture post-translational modifications and hence was used in the following experiments for quantification.

Under basal conditions, ELC was predominantly present in its basal state (Basal: $79.5 \pm 7.1\%$) and a predicted additionally phosphorylated form (+1P: $19.0 \pm 6.5\%$). In response to ionophore stimulation, phosphorylated ELC (+1P: $33.3 \pm 5.9\%$) was significantly increased and a second phosphorylated ELC form (+2P: $8.0 \pm 2.6\%$) became detectable (Fig. 3D). The $Ca^{2+}$-mediated increase of ELC-phosphorylation was blocked by knockdown (KD) of NEK9 (Fold change ELC-phosphorylation; Ctrl: $2.2 \pm 0.3$, KD NEK9: $1.0 \pm 0.1$) (Fig. 3D, E). In addition, NEK9 overexpression (Ox) led to hyperphosphorylated ELC already under basal conditions. In detail, the +1P and +2P phosphorylated ELC forms were significantly increased compared to control (1P + 2P; Ox NEK9:

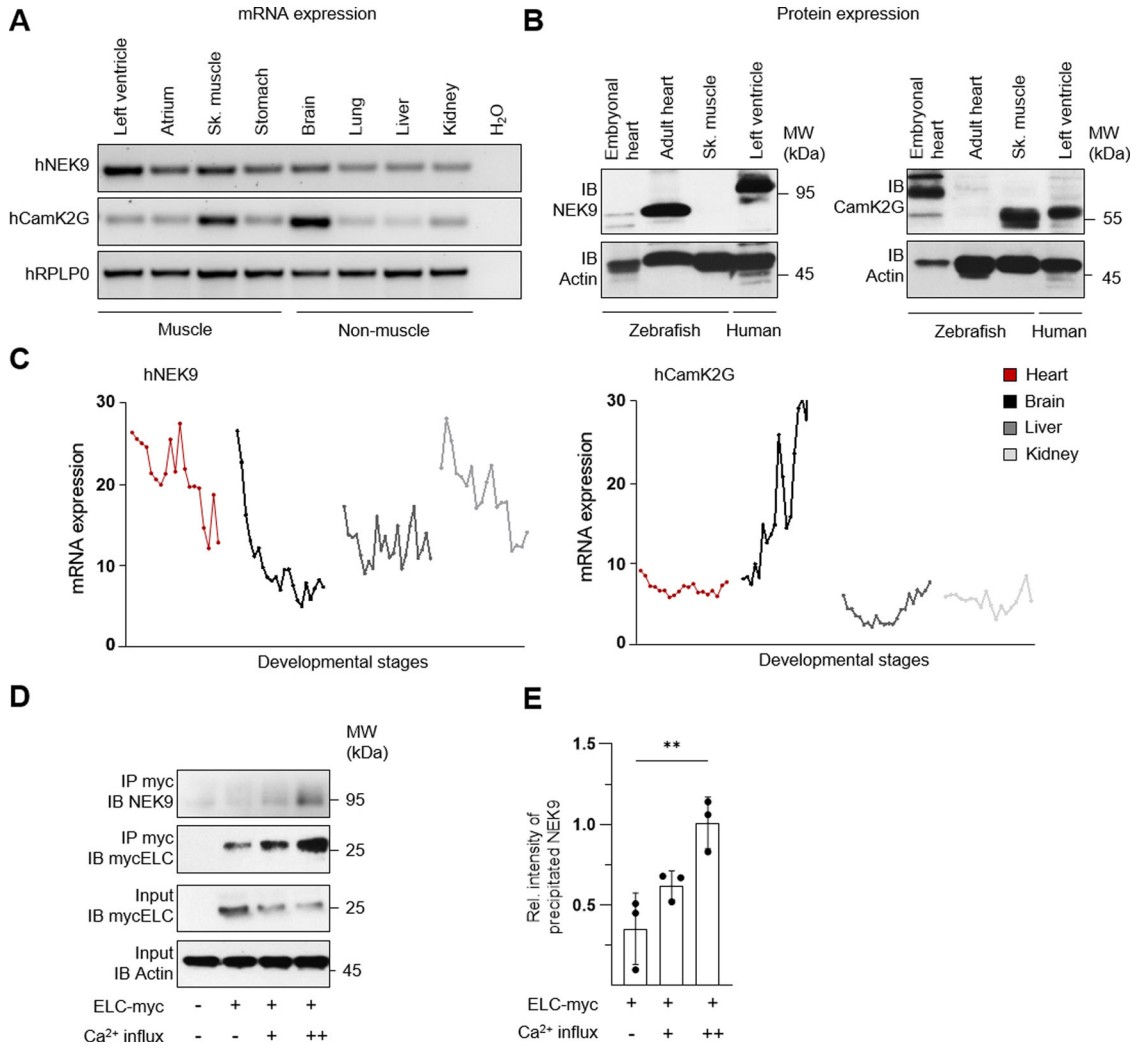

**Fig. 2 | Characterization of ELC interacting kinases. A** NEK9 and CamK2G RNA expression in human total RNA tissue panel. Ribosomal protein large subunit P0 (RPLP0) was used as housekeeping gene. **B** NEK9 (left panel) and CamK2G (right panel) protein expression in piscine and human heart tissue to detect cross-species applicability. **C** mRNA expression profile of NEK9 and CamK2G in different human tissues from 4 weeks post-conception to adulthood[68]. **D, E** Relationship of NEK9-ELC binding efficiency on intracellular $Ca^{2+}$-concentration. NEK9-ELC interaction was analyzed after overexpression of myc-tagged ELC protein in human cells followed by specific immunoprecipitation (IP) and immunoblot (IB) analysis (**D**). $Ca^{2+}$-influx was obtained by ionophore stimulation. Quantification of NEK9-ELC interaction dependent on $Ca^{2+}$-influx illustrated as the averaged intensity of precipitated NEK9 protein normalized to actin protein expression of input protein lysate (**E**). Data are mean ± SD ($n = 3$). **$P < 0.01$ by the analysis of ordinary one-way ANOVA followed by Bonferroni's multiple comparisons test.

48.4 ± 4.9%, Ctrl: 20.5 ± 7.1%, $P < 0.01$) (Fig. 3D, E). $Ca^{2+}$ stimulation did not further increase ELC-phosphorylation in this context (Fig. 3D, E), hinting at a saturation effect of the kinase-target interaction in this experiment. The results indicate (1) a causal relationship between NEK9 activity and ELC-phosphorylation and (2) a stronger interaction of NEK9 and ELC depending on $Ca^{2+}$ levels, resulting in increased ELC-phosphorylation.

## NEK9 and ELC are functionally coupled in the heart

Despite the fact that NEK9 appears to be enriched in human LV tissue, a physiological role of NEK9 in the heart has not been reported so far. To do so, we used zebrafish as model organism and performed (1) transient knockdown by antisense oligonucleotides, (2) CRISPR/Cas9-mediated knockout and (3) genetic sensitizing in an ELC phospho-deficient transgenic zebrafish line. To establish these approaches several prerequisites were considered. First, NEK9 shows high cross-species conservation as shown by immunoblots of heart protein (Fig. 2B) and protein sequence homology alignments (Supplementary Fig. 1A). In addition, pronounced expression of *nek9* in zebrafish hearts can be shown by fluorescence antisense in-situ hybridization

(Supplementary Fig. 3), resembling the findings from human expression data (Fig. 2A). In case of NEK9-knockdown, we observed the development of progressive heart failure with a penetrance of 72 ± 3.5% (Fig. 4A, B) and strongly reduced fractional shortening (FS) of atrium (a) and ventricle (v) at 72 h post fertilization (hpf) (KD-NEK9: aFS: 40.5 ± 2.6%, vFS: 34.4 ± 2.6%; Ctrl: aFS: 60.8 ± 1.3%, vFS: 55.3 ± 0.2%, $P < 0.001$) (Fig. 4C and Supplementary Movie 1). The heart rate was also significantly reduced after NEK9 knockdown at 72 hpf (Supplementary Fig. 4A). The specificity of the knockdown was verified by using a different antisense oligonucleotide targeting the 5'UTR of *nek9* (Supplementary Fig. 4B–F) and performing NEK9 mRNA rescue experiments (Fig. 4B, C). In summary, a specific effect of NEK9 knockdown on cardiac function could be underlined.

Next, we generated transgenic zebrafish using the CRISPR/Cas9 system targeting the NEK9 locus (Fig. 4D–F, Supplementary Fig. 5, and Supplementary Fig. 6). F0 embryos were screened for mutation events at the *nek9* locus by using a heteroduplex assay (Supplementary Fig. 5A). Two different heterozygous transgenic *nek9* lines with different deletion events were identified by two fully informative PCR markers (Supplementary Fig. 5B). In detail, one line is harboring a 78 bp

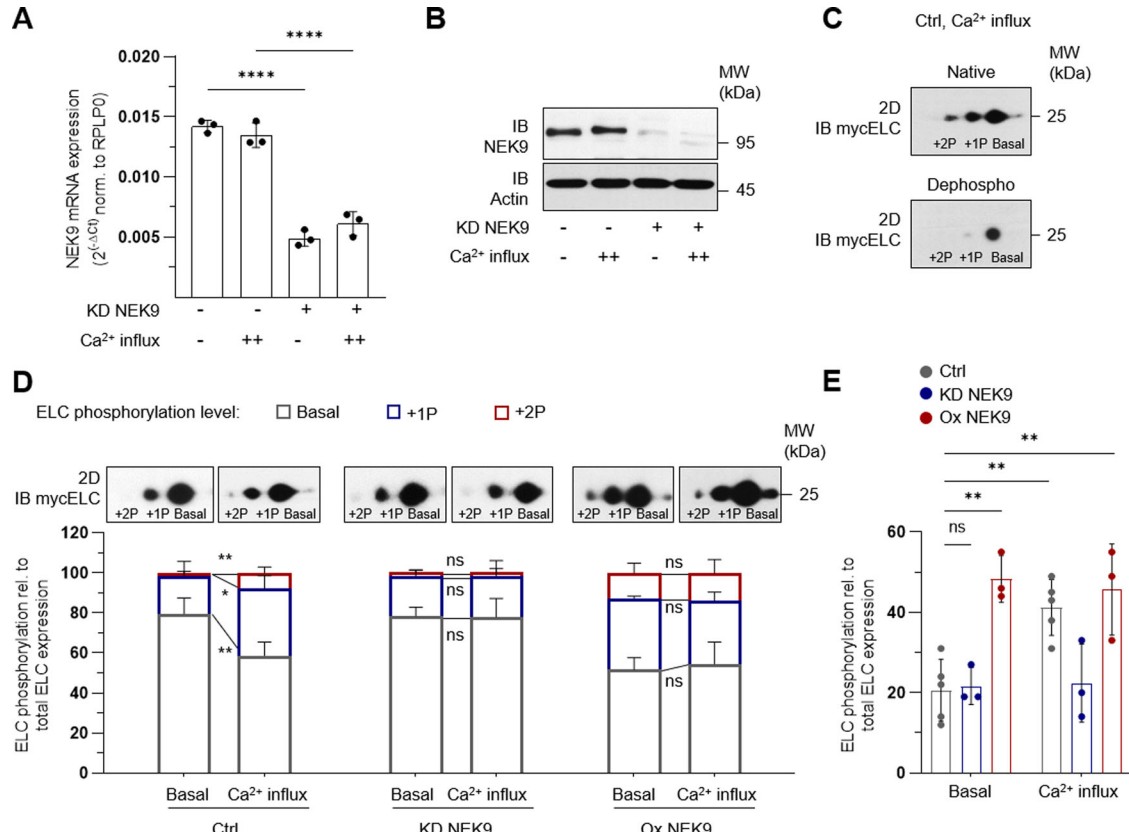

**Fig. 3 | NEK9 regulates ELC-phosphorylation pattern in-vitro. A** Validation of siRNA mediated knockdown (KD) of NEK9 in HEK293 cells by quantitative real time PCR. mRNA expression shown as $2^{(-\Delta Ct)}$ normalized to RPLP0 (ribosomal protein large subunit P0). $Ca^{2+}$-influx was obtained by ionophore stimulation. Data are mean ± SD ($n = 3$). ****$P < 0.0001$ by analysis of ordinary one-way ANOVA followed by Bonferroni's multiple comparisons test. **B** Representative immunoblot (IB) of NEK9 protein expression after specific knockdown (KD). **C** 2D immunoblot of ELC-phosphorylation pattern after myc-tagged ELC overexpression and $Ca^{2+}$-influx using myc-specific antibody. After protein isolation samples were split and half of the sample was treated with phosphatase inhibitor (Native) and the other half was incubated with phosphatase (Dephospho). **D** Representative 2D immunoblot of

myc-tagged ELC-phosphorylation pattern using myc-specific antibody (above) and ELC-phosphorylation illustrated as averaged intensity normalized to total ELC protein (down). ELC-phosphorylation was analyzed after myc-tagged ELC over-expression and siRNA knockdown (KD) of NEK9 or after overexpression (Ox) of NEK9 in human cells. $Ca^{2+}$-influx was obtained by ionophore stimulation. Data are mean ± SD ($n = 3$, $n = 5$ for Ctrl group). *$P < 0.05$, **$P < 0.01$ by the analysis of two-tailed paired Student's t-test. **E** Relative ELC protein phosphorylation (sum of phosphorylated (+1P + 2P) ELC forms) illustrated as averaged intensity normalized to total ELC protein. Data are mean ± SD ($n = 3$, $n = 5$ for Ctrl group). **$P < 0.01$ by the analysis of ordinary one-way ANOVA followed by Bonferroni's multiple comparisons test.

deletion and one line contains a 500 bp deletion of the zebrafish *nek9* gene (Supplementary Fig. 5C). The 78 bp deletion is predicted to lack parts of the protein kinase domain including the ATP binding domain (*nek9*[78del/+]) and the 500 bp deletion results in loss of functional NEK9 protein by a premature stop codon (*nek9*[+/500del]) (Supplementary Fig. 5C and Fig. 5A–C). Compound heterozygous embryos were generated by incrossing of both lines (*nek9*[+/500del] x *nek9*[78del/+]). As shown in Fig. 4E, heterozygous as well as compound heterozygous NEK9 transgenic zebrafish developed heart failure at the embryonic stage. In detail, 38% of heterozygous *nek9*[78del/+] embryos showed impaired cardiac function, around 68% of heterozygous *nek9*[+/500del] embryos developed heart failure and 100% of analyzed compound heterozygous embryos (*nek9*[78del/500del]) presented with the cardiac phenotype at 72 hpf (Fig. 4E). Importantly, the CRISPR/Cas9 mutated embryos resemble the phenotype observed after knockdown of *nek9* (Fig. 4A, D). The heart chambers were dilated and the ventricular walls were slightly thickened due to reduced resorption of cardiac jelly. The looping was still incomplete after 72 hpf. The morphological and functional changes led to reduced blood flow and weaker, but rhythmic contractions of both chambers (Supplementary Movies 1 and 2). The cardiac contractility remains reduced, along with a significantly decreased vFS and aFS at 72 hpf (Fig. 4F). To characterize the two mutated alleles *nek9*[78del] and *nek9*[500del] independently, we incrossed the lines separately within each of them.

Heterozygous embryos of both lines *nek9*[78del/+] and *nek9*[+/500del] developed similar phenotypes (Supplementary Movie 2). As shown in Supplementary Figure 6, genotyping after 24 hpf and 72 hpf indicated embryonic lethality of homozygous embryos (*nek9*[78del/78del] and *nek9*[500del/500del]) and Mendelian ratios were not achieved (Supplementary Fig. 6A). Of note, penetrance was incomplete at 72 hpf (Supplementary Fig. 6B, C). Hence, enough heterozygous embryos survive for line propagation. The expressivity of the heart failure phenotype as reflected by fractional shortening was comparable between both lines (*nek9*[78del/+]: vFS: 32.0 ± 5.1%, *nek9*[+/500del] vFS: 33.1 ± 4.1) (Supplementary Fig. 6B, C). We also overexpressed mutant zebrafish NEK9 protein (NEK9[78del] or NEK9[500del]) in-vitro in combination with zebrafish ELC. NEK9[500del] was not expressed in-vitro, pointing towards non-sense mediated RNA decay or instable partial protein. In contrast, truncated NEK9[78del], which misses a part of the protein kinase domain (illustrated in Fig. 5A) was expressed at equal extent as wild-type NEK9 (Fig. 5B). Of note, using specific IP followed by IB (Fig. 5B), ELC was found to interact less with truncated NEK9[78del] compared to wild-type NEK9 protein, narrowing down the site of interaction to the protein kinase domain. This could be confirmed by several human deletion mutants of NEK9, that were generated by circular mutagenesis. In detail, as shown in Fig. 5D, E, there is a dip of interaction between ELC and NEK9 in case of protein kinase domain deletion or when deleting the

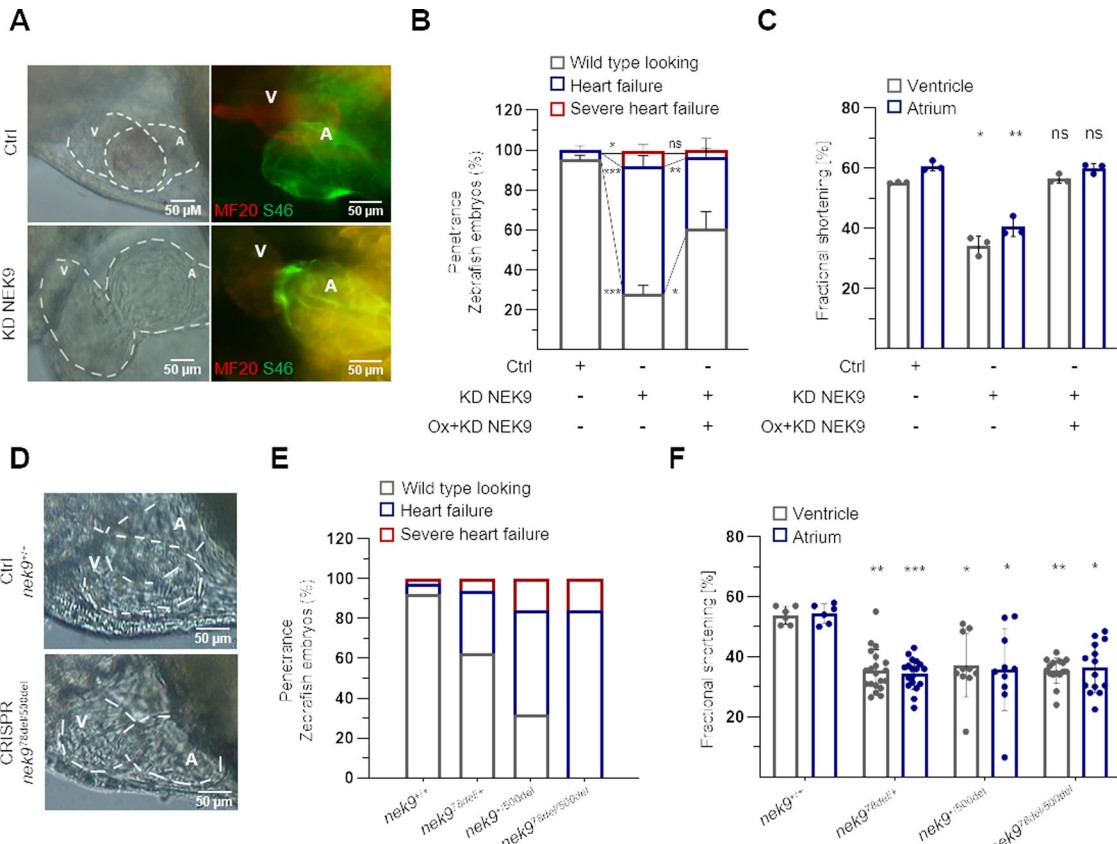

**Fig. 4 | Functional analysis of NEK9 in-vivo. A** Lateral view of zebrafish embryo hearts 72 h post fertilization (hpf) injected with control oligonucleotides or anti-sense oligonucleotides blocking the translational start site of *nek9* (KD NEK9) (left). Immunostaining of atrium-specific (A) and ventricle-specific (V) myosin heavy chains (green: antibody against atrial-specific myosin (S46), red: antibody against ventricular and atrial myosin (MF20)) (right). **B** Phenotype rescue of KD NEK9. Zebrafish eggs were injected with *nek9* mRNA (Ox + KD NEK9) followed by KD NEK9 in one-cell stage. Analysis was performed at 72 hpf. Data are mean ± SD (Ctrl, *n* = 143; KD NEK9, *n* = 192; Ox + KD NEK9, *n* = 155 fish embryos out of 3 experiments). *$P < 0.05$; **$P < 0.01$; ***$P < 0.001$ by the analysis of two-tailed paired Student's t-test. **C** Fractional shortening (FS) of zebrafish morphants at 72 hpf. Data are mean ± SD (Ctrl, *n* = 25; KD NEK9, *n* = 21; Ox + KD NEK9, *n* = 21 fish embryos out of 3 experiments). *$P < 0.05$; **$P < 0.01$ by the analysis of two-way ANOVA followed by Bonferroni's multiple comparisons test. The respective heart rate is shown in

Supplementary Fig. 4A. **D–F** Two heterozygous transgenic zebrafish lines were generated by CRISPR/Cas9. Heterozygous transgenic zebrafish *nek9*[78del/+] and *nek9*[+/500del] (F1 parents) were incrossed and F2 generation was analyzed at 72 hpf regarding heart morphology (**D**), penetrance of heart failure phenotype (**E**) and FS (**F**). Recordings were performed for all embryos showing a heart failure phenotype compared to randomly chosen wild type looking littermates. After phenotype analysis all embryos were genotyped as shown in Supplementary Fig. 5B and genotypes were associated to phenotypes (total number of genotype (number of phenotype: wild type looking/heart failure/severe heart failure): *nek9*[+/+], *n* = 77 (71/4/2); *nek9*[78del/+], *n* = 64 (40/20/4); *nek9*[+/500del], *n* = 19 (6/10/3); *nek9*[78del/500del], *n* = 19 (0/16/3)). Statistical significance in **F** was calculated by a mixed effect model followed by Bonferroni's multiple comparisons test. *$P < 0.05$; **$P < 0.01$; ***$P < 0.001$ are shown for the corresponding chamber of Ctrl embryos.

Regulator of Chromosome Condensation (RCC1, NEK9[Middle(309-726)del]) repeat profile of NEK9.

Next, the causal relationship between NEK9 and ELC-phosphorylation-mediated cardiac contractility was investigated by pathway-directed genetic sensitizing (Fig. 6A) in the ELC phospho-deficient stable mutant zebrafish line *lazy susan* (*laz*[m647])[21,23]. Laz mutants harbor a mutation in the *cmlc1* gene leading to a truncated, but stably expressed ELC protein with loss of carboxy-terminal phosphorylation. Complete loss of the phosphorylation sites in homozygous embryos (*laz*[m647/m647]) is associated with impaired cardiac contractility at rest (*laz*[m647/m647] phenotype), while heterozygous fish (*laz*[m647/+]) are not affected. First, a *nek9* gene dose titration system was established by using wild-type embryos (Fig. 6B). As shown, low dose (LD) of *nek9* antisense oligonucleotides did only result in a small fraction of embryos developing heart failure (LD NEK9 21 ± 4.7%) (Fig. 6B). As such, we noticed a step-wise reduction of NEK9 protein expression with anti-sense dosage (Fig. 6C). Next, cardiac function was studied in the *laz*[m647] embryos after injection of LD *nek9* antisense oligonucleotides. Heterozygous *laz*[m647/+] embryos show strong sensitizing by developing heart failure after 72 hpf, whereas wild-type siblings showed no obvious

phenotype (*laz*[m647/+]: 66 ± 8%; *laz*[+/+]: 19 ± 13%) (Fig. 6E). In detail, LD-*nek9* sensitized *laz*[m647/+] embryos showed appropriate growth of the myo-cardium, but slightly dilated ventricles and atria (Fig. 6D and Supplementary Movie 3). Functionally this was reflected by significantly decreased cardiac contractility of LD-*nek9* sensitized *laz*[m647/+] embryos compared to wild-type (*laz*[+/+]) siblings of the same clutches (vFS: *laz*[m647/+]: 37.3 ± 1.6%; *laz*[+/+]: 51.1 ± 2.9%, $P < 0.001$) (Fig. 6F). Sensitizing with antisense oligonucleotides targeting ELC-independent pathways of cardiac contractility, such as integrin-linked kinase (ILK)[26–28], did not affect heterozygous *laz*[m647/+] embryos (Fig. 6E, F). Thus, sensitizing of *laz*[m647/+] embryos by per se sub-phenotypic *nek9* loss-of-function strongly suggests functional coupling of the NEK9 kinase, ELC-phosphorylation and cardiac contractility.

In conclusion, these experiments underline the role of NEK9 for cardiac development and maintenance of contractility by interacting with ELC via its protein kinase domain.

## ELC is phosphorylated in the human heart
Until now, it was unknown whether the posttranslational ELC-phosphorylation is present or even regulated in the human heart.

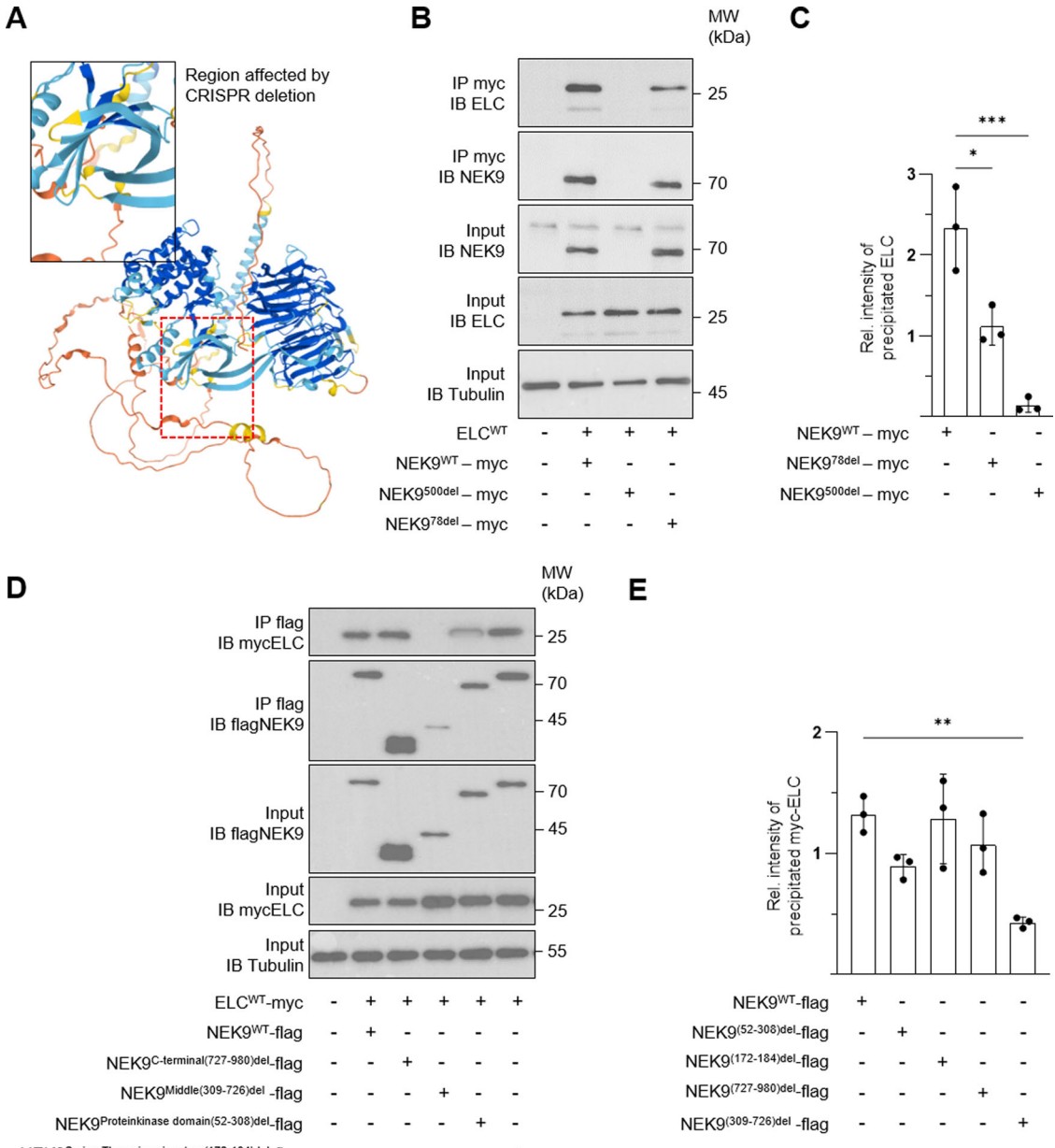

**Fig. 5 | Analysis of the ELC-NEK9 interaction.** Two heterozygous transgenic zebrafish lines were generated by CRISPR/Cas9. **A** NEK9[78del] protein is lacking the ATP-binding domain shown in the 3D structure of NEK9 (**A**). NEK9[500del] is harboring a frame shift mutation leading to a premature stop codon. Sanger sequencing results of the affected allele are shown in Supplementary Fig. 5C. **B**, **C** Piscine ELC and mutated piscine myc-tagged NEK9 protein either lacking the ATP-binding domain (NEK9[78del]) or harboring a premature stop codon (NEK9[500del]) were overexpressed in HEK293 cells. The ELC-NEK9 interaction was analyzed by specific immunoprecipitation (IP) and immunoblot (IB) (**B**). Quantification of ELC-NEK9

interaction illustrated as averaged intensity of precipitated ELC protein normalized to ELC expression of input protein lysate (**C**). Data are mean ± SD ($n = 3$). *$P < 0.05$; ***$P < 0.001$ by the analysis of ordinary one-way ANOVA followed by Bonferroni's multiple comparisons test. **D**, **E** Validation of ELC-NEK9 interaction. Human myc-tagged ELC protein and different deletion variants of human flag-tagged NEK9 protein were overexpressed and interaction intensity was quantified after IP and IB. Data are mean ± SD ($n = 3$). **$P < 0.01$ by the analysis of ordinary one-way ANOVA followed by Bonferroni's multiple comparisons test.

Here, we isolated protein from human left ventricular heart biopsies and separated the proteins by 2D electrophoresis followed by Silver Stain (Fig. 7A). Spot areas were picked and analyzed by LC-MS. vELC and vRLC were identified showing almost full sequence coverage (Fig. 7B). Importantly, atrial isoforms that may interfere with quantification by antibodies had not been detected at the investigated region of interest (Fig. 7B; see also Supplementary Material). Figure 7C presents a map of all detected posttranslational phosphorylations (P*) and deamidations (D*) for vELC and vRLC measured in six independent human left ventricular heart biopsies. We detected nine sites with

phosphorylation in vELC (for corresponding spectra see Supplementary Figure 7). Five of these phosphorylation sites (T72, T88, T127, T129, and S179) could be validated by peptide spectra analysis extracted from Proteomics DB[29,30] and T64, T127, as well as T129, were already described in-vitro by Arrell et al.[19] and Cadete et al.[20]. In contrast to RLC, the ELC-phosphorylation map shows very dense modification events (Fig. 7C).

Using LC-MS, we detected many different protein species sharing the same isoelectric point, making it difficult to analyse their posttranslational modifications by silver staining in one or two dimensions. The spot areas of RLC, e.g. cover three unique protein species, while

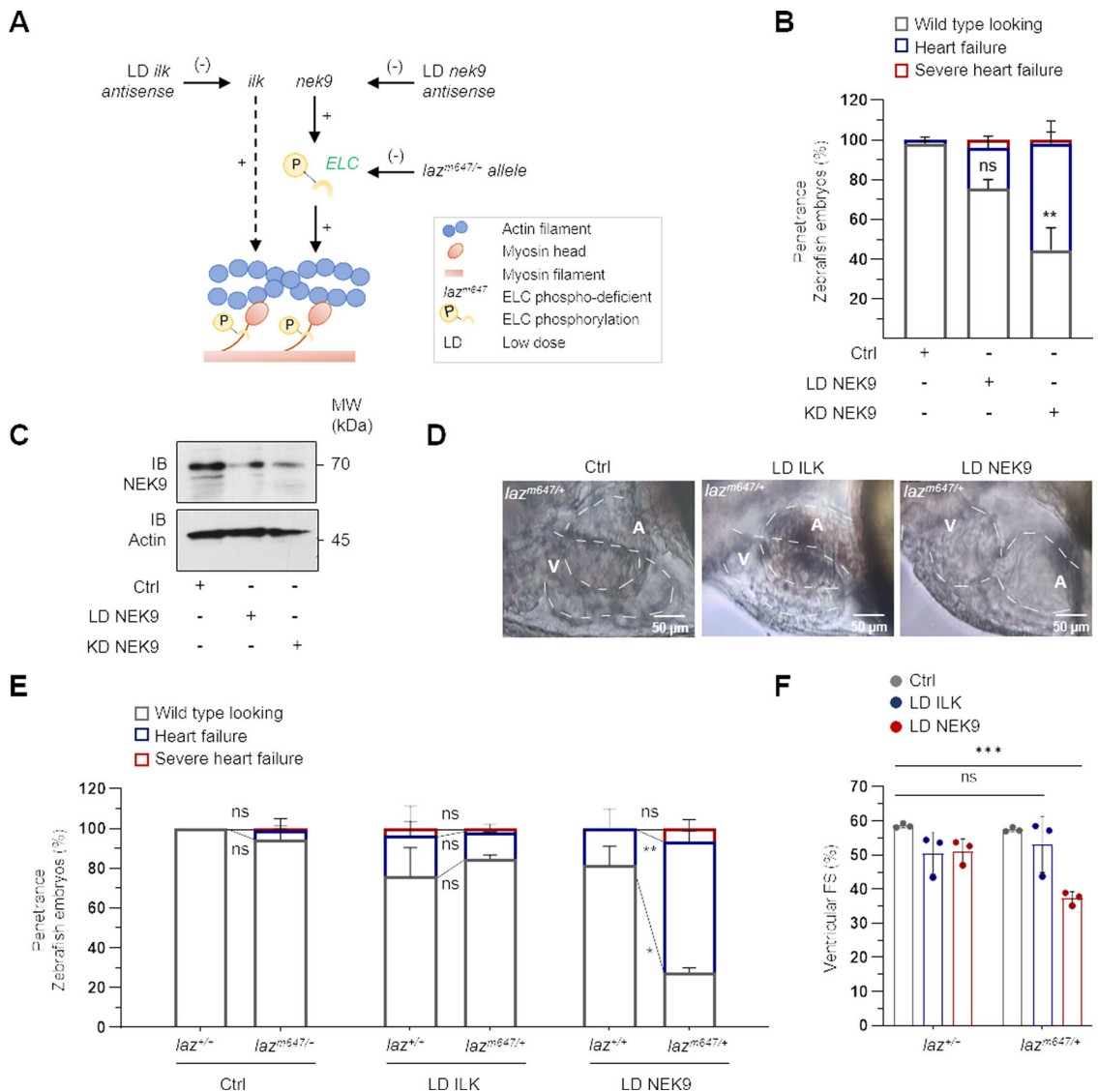

**Fig. 6 | Functional coupling of ELC and NEK9 in-vivo. A** Schematic illustration of genetic sensitizing by *nek9* in an ELC phospho-deficient genetic background. The concept is based on the progressive amplification of a response by functionally coupled factors. **B** Penetrance of NEK9 KD in zebrafish embryos at 72 hpf. Different concentrations of antisense oligonucleotides against zebrafish *nek9* were injected in a genetic wild-type background (KD NEK9: knockdown of NEK9 resulting in a significant heart failure phenotype; LD NEK9: low-dose knockdown showing no phenotypical effect). Data are mean ± SD (Ctrl, *n* = 517; LD NEK9, *n* = 162; KD NEK9, *n* = 461 fish embryos out of 4 (Ctrl), 3 (LD NEK9) and 5 (KD NEK9) experiments). For statistical analysis a mixed effect model followed by Bonferroni's multiple comparisons test was used. **\*\****P* < 0.01 shown for the corresponding embryos injected with Ctrl oligonucleotides. **C** Validation of NEK9 KD or LD NEK9 by immunoblot (IB). **D** Lateral view of heterozygous *lazy susan*

(*laz^{m647/+}*) embryo hearts 72 hpf injected with control oligonucleotides or low-dose antisense oligonucleotides against zebrafish *ilk* (LD ILK) or *nek9* (LD NEK9). **E** Genotype-phenotype-association of *laz^{m647}* zebrafish injected with control oligonucleotides, LD ILK or LD NEK9. Individual genotyping was performed after blinded phenotyping at 72 hpf. Data are mean ± SD (Ctrl: *laz^{+/+}* *n* = 41, *laz^{m647/+}* *n* = 81; LD ILK: *laz^{+/+}* *n* = 21, *laz^{m647/+}* *n* = 46; LD NEK9: *laz^{+/+}* *n* = 49, *laz^{m647/+}* *n* = 97 fish embryos out of 3 experiments). *\*P* < 0.05, *\*\*P* < 0.01 by the analysis of two-tailed paired Student's t-test. **F** Fractional shortening (FS) of genetic sensitized *lazy susan* (*laz^{m647}*) embryos. After analyzing FS, individual genotyping was performed and associated with the phenotype. Data are mean ± SD (Ctrl: *laz^{+/+}* *n* = 41, *laz^{m647/+}* *n* = 82; LD ILK: *laz^{+/+}* *n* = 21, *laz^{m647/+}* *n* = 46; LD NEK9: *laz^{+/+}* *n* = 49, *laz^{m647/+}* *n* = 97 fish embryos out of 3 experiments). *\*\*\*P* < 0.001 by the analysis of ordinary one-way ANOVA followed by Bonferroni's multiple comparisons test.

the ELC region of interest harbors 41 additional proteins (Supplementary Fig. 8). Hence, we developed a 2D-ELC immunoblotting approach for reliable quantification of the human ELC-phosphorylation pattern, which we carefully validated. The ELC antibody was tested for specificity to exclude cross-reactivity (Fig. 7D) and the range of quantification was determined using protein titration experiments and different exposure times (Fig. 7E). In the 2D immunoblot assays, three distinct protein spots were detected for the ventricular ELC protein. Besides the most abundant basic ELC spot, two additional spots shifted towards an acid pI could be observed (Figs. 7F and 8C). To validate that this shift is due to phosphorylation, we

performed in-vitro dephosphorylation assays on human cardiac protein isolates. As shown in Fig. 7F, after treatment with calf intestinal alkaline phosphatase, the acidic-shifted forms of ELC were significantly diminished, signifying that the shifted forms are indeed phosphorylated forms (see also Fig. 8C). In case of RLC, treatment with phosphatase resulted only in a smaller decrease of the shifted form (Fig. 7F), indicative of other posttranslational modifications of RLC, such as the previously described amino acid deamidation of RLC[31]. Activity of the phosphatase and the dephosphorylation protocol was confirmed by immunoblot using phospho-troponin I antibody (Ser23/24) (Fig. 7F) and by performing ProQ experiments (Supplementary Fig. 9) of the

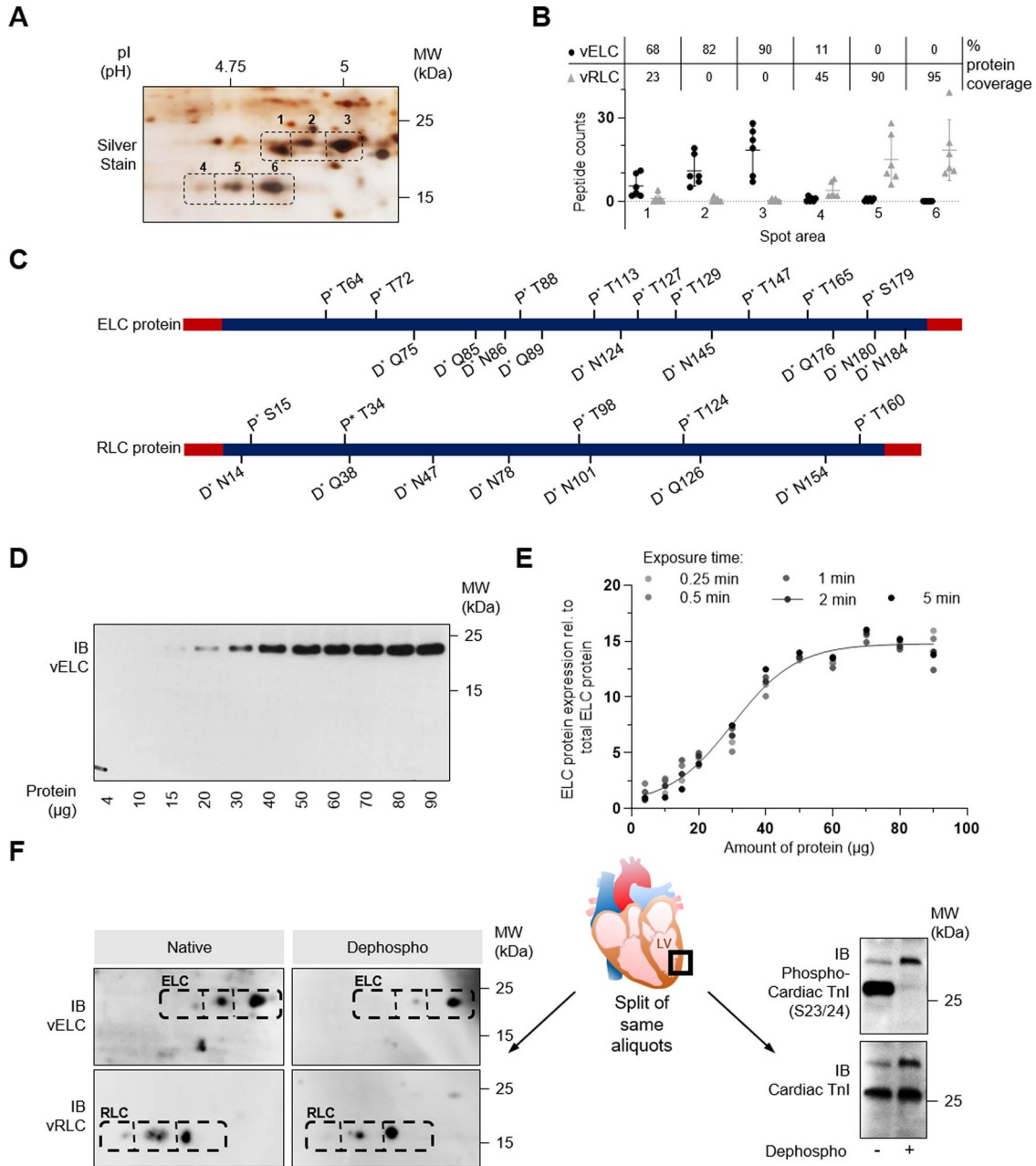

**Fig. 7 | ELC posttranslational modifications in human left ventricular tissue.**
**A** Representative 2D silver stain of human LV tissue. **B** Selected spot areas 1 to 6 were analyzed by liquid chromatography followed by mass spectrometry (LC-MS). Peptide counts uniquely assigned to ventricular (v) ELC or vRLC are shown for 6 independent samples. The mean percentage of the protein coverage of vELC and vRLC are given above (table inset). Data are mean ± SD. **C** Schematic illustration of identified ELC or RLC-phosphorylation sites and deamidated amino acid residues detected in human LV tissue (n = 6; DCM: n = 4; Ctrl: n = 2). **D** Validation of polyclonal rabbit antibody against human ELC (Biozol, GeneTex, ZF127578) by immunoblot (IB) of different amounts of human LV protein. No cross reactivity with vRLC was detected. **E** Quantification of relative ELC protein expression illustrated as averaged intensity normalized to total ELC protein. ELC expression intensities were

plotted against the amount of human heart protein. Exposure time was varied from 0.25 to 5 min. For 2 min exposure $R = 0.9383$ and $P < 0.0001$ was calculated by two-tailed Spearman Rank Correlation (n = 5). **F** Validation of in-vitro dephosphorylation of human LV tissue. After protein isolation samples were split and half of each sample was treated with phosphatase inhibitor (Native), while the other half was incubated with phosphatase (Dephospho). Proteins of the same aliquot were analyzed either by ELC or RLC 2D immunoblot (IB) (left panel) or IB against phospho-cardiac troponin I (TnI) (right panel). The expression of total cardiac TnI was used as loading control. The same aliquots were used for Pro-Q® Diamond Phosphoprotein Stain to detect phosphorylated protein species in 2D IB (Supplementary Fig. 9).

same protein aliquots. Using LC-MS of a representative sample (Supplementary Fig. 10), it could be verified that phosphatase treatment resulted in loss of phosphorylation-status from 63 to 24%. Taken together, we created a human phosphorylation map of ELC and developed a specific 2D-ELC immunoblot assay for quantification of ELC-phosphorylation in human tissue.

## ELC-phosphorylation is increased in human LV tissue of DCM patients

Patients suffering from dilated cardiomyopathy (DCM) and systolic dysfunction undergo adaptive changes in the expression and post-translational modification of various sarcomeric proteins. While in congenital heart disease isoform shifting occurs from ventricular

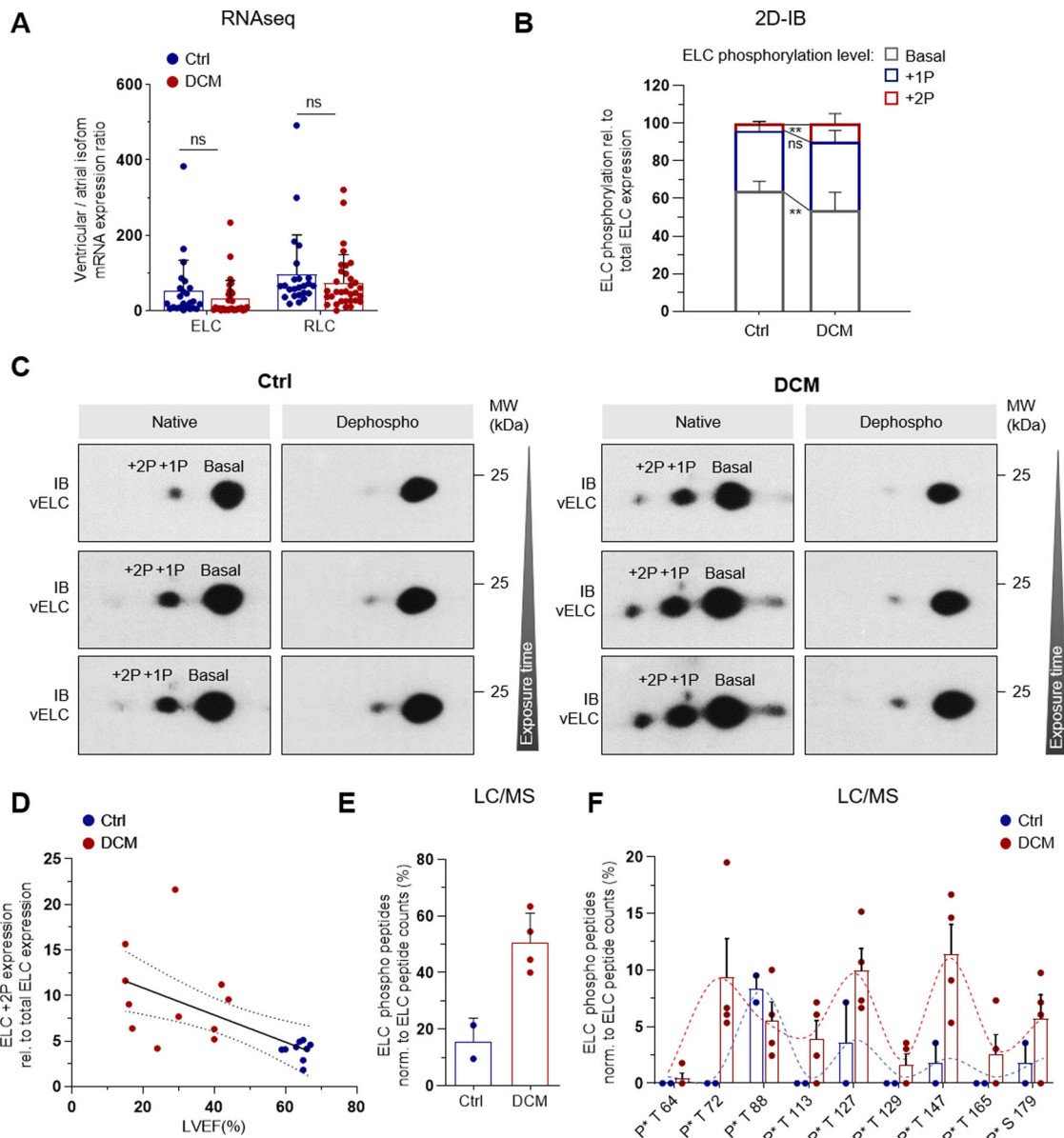

**Fig. 8 | ELC-phosphorylation is regulated in human dilated cardiomyopathy.**
**A** Ratio of ventricular (v) MLC and atrial (a) MLC mRNA expression in human left ventricular tissue of DCM patients and HTX controls (HTX: *n* = 28; DCM: *n* = 44). Samples were analyzed by deep mRNA sequencing (mRNAseq)[38]. Data are mean ± SEM. Statistical significance was tested by two-tailed unpaired Student's t-test. **B** Quantification of relative basal (Basal) and phosphorylated (+1P, + 2P) ELC protein illustrated as averaged intensity normalized to total ELC protein (Control: *n* = 9; DCM: *n* = 11). 6 HTX controls (Supplementary Table 1) and 3 healthy controls were included. Data are mean ± SD. **P < 0.01 by the analysis of two-tailed unpaired Student's t-test. **C** Representative ELC 2D immunoblot (IB) of human LV heart tissue of a DCM patient and a healthy control. Native ELC protein forms (left panel) and ELC protein forms after in-vitro dephosphorylation (right

panel). Exposure times were varied from 1 to 5 min. **D** Correlation analysis of phosphorylated ELC expression (+2P) and left ventricular ejection fraction (LVEF). *R* = −0.6193 and *P* = 0.0036 by the analysis of two-tailed Pearson Correlation (Control: *n* = 9; DCM: *n* = 11). **E** Sum of phospho peptides detected in spot arrays assigned to ventricular (v) ELC (see Fig. 7A, B) normalized to total ELC peptide counts. Proteins were separated by 2D electrophoresis and selected spot areas were analyzed by liquid chromatography followed by mass spectrometry (LC-MS) (DCM: *n* = 4; Ctrl: *n* = 2). Data are mean ± SD and are shown without statistical analysis. **F** Distribution of ELC-phosphorylation in DCM patients compared to healthy controls (Ctrl). Normalized amount of ELC phospho-peptides shown for the respective ELC-phosphorylation site (DCM: *n* = 4; Ctrl: *n* = 2). Data are mean ± SEM and are shown without statistical analysis.

(vELC) to atrial myosin light chain (aELC) expression[11], analysis of a DCM cohort by mRNA-seq did not show significant isoform expression changes of ELC (Fig. 8A). Hence, we focused on post-translational ELC modification as a potential contributor to adaptive or maladaptive changes in ELC function.

To quantify ELC-phosphorylation, total protein was isolated from left ventricular (LV) biopsies of 11 DCM patients and 9 control samples (DCM: *n* = 11, HTX: *n* = 6, healthy controls: *n* = 3). The patients suffering from DCM had a severely reduced LV ejection fraction (LVEF) compared to controls (Control: 63.2 ± 3.2%; DCM: 28.4 ± 11.7%; *P* < 0.0001)

(Supplementary Table 1). Ventricular ELC protein isoforms were separated as described in Fig. 7 and quantified by densitometry (Fig. 8B). As our method cannot distinguish, which residue is phosphorylated, ELC protein isoforms were assigned to basal (B), +1P and +2P phosphorylated forms. The +2P ELC protein level was significantly increased in DCM patients (DCM: 9.8 ± 4.9%; Control: 4.0 ± 1.0%; *P* = 0.0034) going along with a decrease in basal ELC protein (Fig. 8B, C). Interestingly, +2P ELC protein level of all samples negatively correlated with LVEF (*R* = −0.6193 and *P* = 0.0036) (Fig. 8D) and the level of +1P and +2P phosphorylated ELC forms is altogether increased in

DCM hearts (DCM: 46.3 ± 9.0%; Control: 35.9 ± 4.7%, $P = 0.0086$). In a validation cohort of 4 DCM patients and 2 healthy control samples we performed LC-MS and analyzed the detected phospho peptides semiquantitativly (Fig. 8E and Supplementary Fig. 10). We could confirm an increase of detected ELC phospho-peptides in DCM patients compared to controls normalized to the total ELC peptide counts (Fig. 8E). Visualization of the abundance of ELC phospho peptides along the ELC protein indicates that phosphorylations are accumulated around certain residues (T72, T88, T127 and T147) (Fig. 8F). Not all of these 4 phosphorylations, however, need to be present in each ELC molecule at a time. To conclude, in early disease stages of DCM (NYHA I: 45%; NYHA II: 36%, NYHA III: 18%) the ELC-phosphorylation level is upregulated, indicating the relevance of ELC-phosphorylation for the adaptation of sarcomere function under pathophysiologically increased cardiac demands.

## Discussion

Finetuning of sarcomere function depends on many factors, including calcium ($Ca^{2+}$) -influx, myosin isoform expression, and phosphorylation of sarcomere proteins. Although ELC is shown to be thiophosphorylated by MLCK in-vitro[32,33], the kinases and phosphatases responsible for regulating ELC-phosphorylation under physiological conditions and in-vivo remained largely elusive. Here, we revealed that NIMA-related kinase 9 (NEK9), which is highly expressed in cardiac tissue, interacts with ELC in a $Ca^{2+}$-dependent manner and regulates its phosphorylation in the heart. Loss of NEK9 is associated with heart failure and causally linked to ELC and cardiac contractility regulation.

NEK9 belongs to the NIMA ('never in mitosis gene A') serine/threonine kinase protein family and is associated with cell cycle control during mitosis[34,35]. NEK9 is found to be highly upregulated in cancerous lung epithelial cells. Furthermore, depletion of NEK9 results in inhibition of cell-cycle progression and proliferation of cancer cells[36]. A homozygous stop-gain mutation in NEK9 is associated with lethal skeletal muscle dysplasia, hinting at a role of NEK9 in muscle tissue. Functionally, fibroblasts of patients show a complete loss of NEK9 protein leading to reduced cell proliferation and a delay in cell cycle progression[37]. Interestingly, we detected enriched NEK9 expression in human LV tissue and a high expression throughout human heart development. Compared to CamK2G, NEK9 shows a 10-fold higher expression in left ventricular tissue on the mRNA level (CamK2G: 0.79 ± 0.12 TPM; NEK9: 9.79 ± 1.02 TPM)[38], which could be recapitulated in zebrafish. In-silico kinase-specific phosphorylation prediction for ELC ranked NIMA-related kinases as the most likely candidates[39].

Ionophore stimulation of human cells signified that the interaction of NEK9 and ELC can be mediated by increased intracellular $Ca^{2+}$ -concentrations, which is the main trigger of cardiac contraction coupling and which is different from smooth muscle cell contraction. Since NEK9 does not seem to have EF-hand motifs, $Ca^{2+}$ might rather improve the binding likelihood of NEK9 and ELC as shown for other protein-protein interactions, potentially also including additional mediators[40,41]. We postulate increased ELC-phosphorylation with increased interaction of ELC with NEK9, which is blunted after loss of NEK9 in $Ca^{2+}$-saturated cells. By characterization of the interaction regions of NEK9 with ELC, we could identify the protein kinase domain as relevant part of NEK9 in two species, human and zebrafish. Other interactors are likely, e.g. by the NEK9 RCC domain, which is known to interact with Ran GTPase regulating mitotic progression[42].

Our in-vitro data corroborate in-vivo studies in transgenic zebrafish lacking the carboxy-terminal ELC-phosphorylation site, where homozygous founders die during development and adult heterozygous fish exhibit progressive deterioration of cardiac performance in response to physical stress[23]. KD of the identified ELC kinase NEK9 results in a significant decrease of ELC-phosphorylation at baseline and also blunting of $Ca^{2+}$ mediated phosphorylation increase.

To the best of our knowledge, NEK9 was previously not described in cardiac pathophysiology. However, NIMA-related kinase 6, another member of the NIMA kinase family, was analyzed in a mice model of cardiac hypertrophy[43]. The absence of NEK6 promotes the progression of cardiac hypertrophy by activation of the Akt signaling pathway. In addition, elevated NEK6 protein levels are detected after transaortic constriction in murine heart tissue and in end-stage patients suffering from dilated cardiomyopathy[43]. Here, we observed that the transient or genetic loss of *nek9* leads to heart failure and severely impaired cardiac contractility in zebrafish embryos. Transgenic *nek9* zebrafish lines lacking the protein kinase domain develop the same heart failure phenotype indicating an essential role of NEK9 for cardiac function. In addition to the induction of heart failure, the mutant *nek9* alleles lead to early embryonic degradation. This is in contrast to transgenic zebrafish lines harboring mutations in cardiac restricted genes, like *lazy susan*[21], *flatline*[44], or *bungee*[45], which compensate a cardiac dysfunction up to day 5 post fertilization by oxygen diffusion via the surrounding medium. The lethal nature of the *nek9* mutation might be explained by the expression of NEK9 in non-cardiac tissues (Fig. 2A) and a broader role of NEK9 during embryogenesis[34,37]. While homozygous NEK9 mutant embryos die in early embryonic stages, combined heterozygous embryos (*nek9[78del/500del]*) show lower lethality and could be characterized at 72 hpf. Heterozygous embryos (*nek9[78del/+]* and *nek9[+/500del]*) can be occasionally raised to adulthood if heart failure is not severe and are available for line propagation.

In the present study, we focused on the identification of NEK9 regulating ELC-phosphorylation in the heart, since ELC is a key player of force transmission in cardiomyocytes. The relevance of ELC-phosphorylation for cardiac contractility was previously shown in zebrafish hearts in-vivo. Complete loss of a C-terminal ELC-phosphorylation site in a stable mutant zebrafish line *(lazy susan)* is associated with severely impaired cardiac contractility of both chambers. While the ELC mutation is lethal in homozygous fish embryos, heterozygous mutant embryos do not show obvious spontaneous phenotype and have a normal life span[21]. Using this model in a heterozygous state for genetic-sensitizing, hypomorphic *nek9* leads to the phenocopy of the homozygous loss of C-terminal ELC-phosphorylation, underpinning functional coupling of NEK9 kinase and ELC.

ELC-phosphorylation has sparsely been reported previously. This might be due to the fact that many other protein species are accumulated in the region of ELC on 2D electrophoresis. We have overcome this issue by coupling LC-MS and specific immunoblotting assays. We could identify as much as nine active phosphorylation sites of ELC, from which we could validate five in additional peptide databases. Since phosphorylation events in bottom-up MS-experiments are close to modifications such as sulfonation (same charge, almost same mass)[46], it was important to combine the assays with dephosphorylation protocols that have been meticulously validated in this study. It is not possible to verify which of the two mainly detected 2D-IB spots are originating from these residues[47], since they might not be phosphorylated in the same ELC molecule at the same time or with differing abundance. Hence, phospho-specific antibodies for the four mainly differing serine/threonine phosphorylation-sites (or even all nine sites) need to be generated for future studies.

The functional relevance of RLC-phosphorylation is demonstrated by the expression of pseudo-phosphorylated RLC protein leading to reconstitution of inotropy in rodent models of heart failure[48–50]. In addition, previous studies on RLC suggest decreased RLC-phosphorylation level in ventricles of end-stage heart failure patients[51–53] and mouse models for cardiac dysfunction[54,55]. Here, we demonstrate the presence of phosphorylation of ELC in human hearts and observed significantly upregulated ELC-phosphorylation in LV tissue of DCM patients, showing an inverse correlation with ventricular ejection fraction. This might indicate a diverging role of RLC and ELC during the pathogenesis of heart failure, or it could be stage

dependent, since we included living DCM patients in earlier disease stages (NYHA I: 45%; NYHA II: 36%, NYHA III: 18%).

We hypothesize that the increase in ELC-phosphorylation serves as an adaptive mechanism in DCM, potentially due to increased ß-adrenergic stress and/or increased $Ca^{2+}$-load. Hence, ELC-phosphorylation may be up-regulated to salvage the circulatory supply/demand mismatch. Mechanistically, ELC-phosphorylation appears to result in enhanced actin-myosin cross-bridge formation and increased force development[23] with RLC- and ELC-phosphorylation being increased after physical stress in the zebrafish heart[23]. In heart failure, intracellular $Ca^{2+}$-levels are increased, which might stimulate the ELC kinase NEK9 as shown in this study in-vitro. This discovery could be important for targeting the sarcomere as a modulator of inotropy, a treatment strategy with high clinical potential[56].

In summary, we identified NEK9 as a bona fide ELC kinase, regulating ELC posttranslational phosphorylation in a $Ca^{2+}$-dependent manner. By demonstrating elevated ELC-phosphorylation levels in DCM patients, important insights could be provided, combining posttranslational ELC modification and human heart disease. Enhancing ELC-phosphorylation might be a mechanism that positively affects cardiac output and hence might be a potential pharmacological target.

## Methods

### Human left ventricular heart tissue
The characterization of samples and patient data within a central biobank approach has been approved by the ethics committee, medical faculty of Heidelberg (S390-2011). The participants have given written informed consent. A prerequisite for enrollment was leftover myocardial tissue from the routine diagnostic workflow. Altogether, remaining tissue biomaterial of 11 DCM patients was analyzed. Further inclusion criteria for DCM patients were defined as left ventricular ejection fraction (LVEF) of lower than 45% and availability of physical activity NYHA classification of the patients as well as N-terminal pro-hormone of brain natriuretic peptide (NT-proBNP) levels and high sensitivity measurement of troponin T (hs-TnT) levels. For controls 6 biopsies of patients after successful heart transplantation (HTX), who underwent routinely pathological check, were used. Their LVEF was defined as at least 55% or higher with good physical activity (NYHA I) and hs-TnT lower than 51 ng/L. Biopsies were obtained from the apical part of the left ventricular wall from patients undergoing cardiac catheterization using a standardized protocol[38,57]. Biopsies were rinsed with NaCl (0.9%) and immediately transferred and stored in liquid nitrogen until protein isolation. Besides the 6 HTX controls, 3 healthy control samples were included and have been used according to the protected health information (45 C.F.R. 164.514 e2) (bioserve) as well as the BCI informed consent F-641-5 (biochain).

### Protein isolation from human left ventricular heart tissue
For protein extraction human biopsies were homogenized by peqlab mill (program: 2 × 5000 rpm 20 s with 10 s pause) in lysis buffer (in mM): 50 Tris-HCl, pH 7.5, 120 NaCl, 5 EDTA, 0.1% NP-40, 1 DTT, 1 NaVO$_3$, 1 NF, 0.2 PMSF, protease inhibitor (cOmplete™ Protease Inhibitor Cocktail, Roche #04693116001) and phosphatase inhibitor (EasyPACK tablet, Roche #4906845001). All steps were performed at 4 °C to avoid native dephosphorylation of the protein extract. The concentration of the supernatant was measured using Bradford Assay (Coomassie Protein Assay Kit, Pierce™ #23200). 45 µg protein of each sample was precipitated for 45 min at 4 °C with one volume 10% TCA. Pellets were washed twice with ice-cold acetone and stored in acetone at −80 °C until use.

### In-vitro dephosphorylation of proteins
For in-vitro dephosphorylation assays, protein was isolated in EDTA-free lysis buffer without addition of phosphatase inhibitor (in mM): 50 Tris-HCl, pH 8, 100 NaCl, 10 MgCl$_2$, 1% NP-40, 1 DTT, protease inhibitor

(cOmplete™ EDTA-free Protease Inhibitor Cocktail, Roche # 4693132001). One half of each sample was treated with phosphatase inhibitor (EasyPACK tablet, Roche #4906845001). The other half was incubated with calf intestine phosphatase (Sigma, #P4978) for 1 h at 37 °C. 45 µg protein of each sample was precipitated for 45 min at 4 °C with one volume 10% TCA. Pellets were washed twice with ice-cold acetone and stored in acetone at −80 °C until use.

### 2D gel electrophoresis
Dried protein pellets were dissolved in 2D sample buffer (in mM): 7000 urea, 2000 thiourea, 4% Chaps, 1% TritonX-100, 32.4 DTT, carrier ampholytes (Bio-Lyte® 3/10 Ampholyte Biorad #163-2094); at 37 °C. The PROTEAN® IEF System from Biorad and 17 cm IPG strips pH 4-7 (IPG ReadyStrip™ Biorad #163-2008) were used for isoelectric focusing (IEF). The strips were allowed to rehydrate overnight and focused as described (PROTEAN® IEF Cell Instruction Manual). After IEF each strip was equilibrated in DTT and IAA equilibration buffer (in mM): 50 Tris pH 8.8, 6000 urea, 33% glycerol, 4% SDS, EB1: 130 DTT, EB2: 162 IAA; for 20 min. Half of each strip (pH 4-5.3, ca. 7.5 cm) was opposed to the second dimension gel (hand-cast 12% polyacrylamide gel, Mini-PRO-TEAN® Cell Biorad) and run for 1 h at 7 mA per gel and 3.5 h at 15 mA per gel in Tris-Glycine buffer. A molecular weight marker (PageRuler™ prestained, ThermoFisher #26617) was moved into a small piece of filter paper and placed in front of the IPG strip before the gel run. For some validation experiments an additional marker was placed at the end of the strip (Precision Plus Protein™ WesternC™ Blotting Standards, Biorad #1610385). 2D separated gels were either used for protein transfer to PVDF membranes and immunostaining or high-sensitive silver staining followed by cutting of protein-spots and mass spectrometry analysis.

### Silver staining
Staining of proteins in polyacrylamide gels was performed by the *Silver Stain for Mass Spectrometry Kit* (Pierce, #24600). Indicated spots were cut out and de-stained according to manufacturer's protocol.

### Phosphoprotein stain
2D polyacrylamide gels were stained for 1 h with Pro-Q® Diamond Phosphoprotein Stain (Invitrogen, #P33300). Fixation, washing, and destaining were performed according to the manufacturer's guidelines. For total protein stain gels were stained overnight with SYPRO® Ruby (Invitrogen, #S12001).

### Antibodies
All primary antibodies used in this study are: rabbit polyclonal antibody against human ELC (Biozol, GeneTex, ZF127578), custom-made rabbit polyclonal antibody against piscine ELC (Eurogentec, immunization peptide: NH2-CAPVPETPKEPEVDLK-CONH2), mouse polyclonal antibody raised against the full-length protein of human CamK2G (MaxPab, Abnova, H00000818), mouse monoclonal antibody against human NEK9 (Abgent, AT3019a, clone name 1F6), mouse monoclonal antibody against actin (Sigma, #A2172, clone name 5C5), mouse monoclonal antibody against beta-actin (Sigma, #A5441, clone name AC-15), mouse monoclonal antibody against beta-actin (Cell signaling, #3700, clone name 8H10D10), mouse monoclonal antibody against flag-tag (Sigma, #F3165, clone name M2), rabbit polyclonal antibody against beta-tubulin (Abcam, #ab6046), rabbit polyclonal antibody against troponin I (Cell signaling #4002) and rabbit polyclonal antibody against phospho-troponin I (cardiac) (Ser23/24) (Cell signaling #4004). HRP-coupled secondary antibodies (Cell signaling, anti-mouse #7076, anti-rabbit #7074) and HRP-conjugated monoclonal antibody against myc-tag (Cell signaling, #2040, clone name 9B11) were used. Antibody dilutions are provided within the Source Data file.

## ELC-specific affinity assay

For ELC-specific co-immunoprecipitation the Dynabeads Co-immunoprecipitation Kit (Invitrogen, #14321D) in combination with Dynabeads™ M-270 Epoxy (Invitrogen, #14302D) was used. Custom specific ELC antibodies (Eurogentec) against zebrafish ELC were covalently coupled by the Dynabeads Antibody Coupling Kit (life technologies, #14311D). All steps were carried out according to the manufacturer's instructions. Precipitated proteins were separated by polyacrylamide gels and stained with custom-made rabbit polyclonal antibody against piscine ELC (Eurogentec, immunization peptide: NH2-CAPVPETPKEPEVDLK-CONH2) for validation of successful co-immunoprecipitation. One representative experiment was sent for mass spectrometry analysis. To do so, the separated proteins were stained with Quick Coomassie® Stain (Serva, #35081) according to the manufacturer's instructions. For every sample three gel pieces were cut out (1.0 ≥ 75 kDa; 2.75–35 kDa; 3.0 ≤ 35 kDa) and liquid chromatography followed by mass spectrometry (LC-MS) analysis was performed. After peptide identification Scaffold (version Scaffold_4.8.9, Proteome Software Inc., Portland, OR) was used to visualize MS/MS based peptide and protein identifications. The identified exclusive peptide counts were summarized for each protein. Peptide identifications were accepted if they could be established at greater than 95.0% probability by the Peptide Prophet algorithm[58]. Protein identifications were accepted if they could be established at greater than 95% probability and contained at least 2 identified peptides. Protein probabilities were assigned by the Protein Prophet algorithm[59]. Specific interaction partners were identified by calculating the foldchange of unique peptide counts enriched by ELC compared to isotype control (IgG Ctrl). The identified proteins were categorized manually: a, sarcomeric proteins; b, enzymes; c, cytoskeleton elements; d, kinases and phosphatases; e, others; f, uncharacterized. Uncharacterized proteins refer to predicted but uncharacterized proteins in the data base. Finally, the proteins were ranked by descending foldchange expecting high abundant proteins to be candidates for further validation.

## Liquid chromatography mass spectrometry (LC-MS)

Separated proteins were processed as described by Fecher-Trost et al. with minor modifications[58]. In brief, after reduction with dithiothreitol and alkylation with iodoacetamide trypsin digestion was done overnight at 37 °C. The reaction was quenched by addition of 20 µL of 0.1% trifluoroacetic acid (TFA; Biosolve, Valkenswaard, Netherlands) and the supernatant was dried in a vacuum concentrator before LC-MS analysis. Nanoflow LC-MS2 analysis was performed with an Ultimate 3000 liquid chromatography system coupled to an QExactive HF mass spectrometer (Thermo-Fischer, Bremen, Germany). Dried samples were dissolved in 0.1% TFA and loaded on a C18 Acclaim PepMap100 trap-column (Thermo Fisher Scientific) with a flow rate of 30 µL/min 0.1% TFA. Peptides were eluted and separated on an C18 Acclaim PepMap RSLC analytical column (75 µm × 250 mm, Thermo Fisher Scientific) with a flow rate of 300 nL/min in a 60 min gradient of 3% buffer A (0.1% formic acid) to 40% buffer B (0.1% formic acid/acetonitrile). The mass spectrometer was operated in data-dependent acquisition mode, automatically switching between MS and MS2. Collision induced dissociation MS2 spectra were generated for up to 20 precursors with normalized collision energy of 29%. Dynamic exclusion was set to 10 s. Raw LC-MS files were processed using Proteome Discoverer 2.3 with SEQUEST HT: ISE (2.0.0.24, ×64) (Thermo Scientific) for peptide identification and quantification. MS2 spectra were searched with the SEQUEST software (Thermo Scientific) against the Uniprot human proteome database (UP000005640_9606) and the contaminants database (MaxQuant version 1.5.3.30; 142525 database entries)[59] with the following parameters: Carbamidomethylation of cysteine residues as fixed modification and Acetyl (Protein N-term),

Oxidation (M), deamidation (NQ) and phosphorylation (S,T,Y) as variable modifications, trypsin/P as the proteolytic enzyme with up to 2 missed cleavages. Target FDR for the total number of identified peptide spectra matched to a protein (PSM) was set to 0.01 and results were filtered for high confident peptides. Peptide identifications were accepted if they could be established at greater than 95.0% probability by the Peptide Prophet algorithm[60]. Protein identifications were accepted if they could be established at greater than 95% probability and contained at least 2 identified peptides. Protein probabilities were assigned by the Protein Prophet algorithm[61]. Peptides of the same sequence with different modifications were counted as different peptides. The assignment of all fragment spectra of peptides labeled as phosphorylated by database search were manually verified. No additional algorithm for posttranslational modification localization had to be used. Due to the high degree of neutral loss (−80 Da, HPO₃), phosphorylation at tyrosine was excluded.

## Human tissue RNA panel

Ventricle (left, R1234138-50), atrium (left, R1234126-50), skeletal muscle (R1234171-50), stomach (R1234248-50), brain (R1234035-50), lung (R1234152-50), liver (R1234149-50) and kidney (R1234142-50) were purchased at Amsbio LLC, UK, quantified with the Denovix DS-11 spectrophotometer and 1000 ng were transcribed into cDNA.

## Quantitative real-time PCR

Reverse Transcription was performed by MuLV Reverse Transcriptase (50 U/µL, ThermoFisher, #N808-0018) in combination with: 25 mM MgCl2 solution and 10x PCR Buffer II (ThermoFisher, #N808-0010), dNTP Mix (10 mM each, ThermoFisher, #R0192), Random Hexamer Primer (100 µM, ThermoFisher, #S0142) as well as RNase Inhibitor (100 µM, ThermoFisher, #E00381). For quantitative real time-PCR SYBR® Green PCR Master Mix (Life Technologies, #4309155) in combination with the ViiA 7 Real-Time PCR System (Applied Biosystems) was used according to the manufacturer's instructions. The primer sequences against human genes were as follows: *nek9*-F: 5′ GTGGTC ACAGTGGAGAAGGA 3′, *nek9*-R: 5′ GCTGTCGATAGGAGGCTTTG 3′, *camk2g*-F: 5′ GTGAAGAAAACCTCCACGCA 3′, *camk2g*-R: 5′ ATCC-GAGCCTCACGTTCTAG 3′, *rplp0*-F: 5′ TCGACAATGGCAGCATCTAC 3′, *rplp0*-R: 5′ ATCCGTCTCCACAGACAAGG 3′. PCR amplificates were separated by agarose gel electrophoresis.

## Cell culture

HEK 293A (Life technologies, #R70507) cells were cultured in DMEM (Gibco, #21969-035) supplemented with 10% FCS (Merck, #S0115), 1% Penicillin-Streptomycin (10,000 U/mL, Gibco, #15140-122) and 1% glutamine (Gibco, #25030-024) at 37 °C, 5% $CO_2$.

## Plasmid overexpression in HEK293 cells

ELC and NEK9 were amplified from human left ventricular cDNA[38] or zebrafish cDNA of the different transgenic mutant *nek9* lines using primers flanking the coding sequence. PCR reaction was performed using KOD Hot Start DNA Polymerase (Merck, # 71086). Cloning of the PCR amplicons was obtained into the mammalian expression vector pcDNA3.1™ (+) (ThermoFisher, #V79020) with N-terminal myc for human ELC (Figs. 1E, 2D, 3C–E; Supplementary Figs. 2, 7) or flag tag for piscine NEK9 using In-Fusion® HD Cloning Plus kit (Takara Clontech, #638910) according to manufacturer's instruction. For the human NEK9 deletion mutants the NEK9-pcDNA3.1 plasmid was used as PCR template harboring a flag tag and variants were generated by the In-Fusion® Snap Assembly Master Mix (Takara Clontech, #638947) according to the manufacturer's instruction. Plasmid sequences were checked by Sanger sequencing before use. For overexpression in HEK293 cells, $2 \times 10^6$ cells were seeded in 150 mm dishes in DMEM media without antibiotics and cultured overnight. 2 µg plasmid were

used for transfection with lipofectamine (Invitrogen, #13778-150). Cells were stimulated or harvested 48 h post transfection.

### siRNA mediated knockdown in HEK293 cells

For siRNA mediated knockdown in HEK293 cells, $2 \times 10^6$ cells were seeded in 150 mm dishes in DMEM media without antibiotics and cultured overnight. After 24 h cells were transfected using lipofectamine (Invitrogen, #13778-150). The siRNA against human NEK9; sih-NEK9: 5′ AAUUGAUCCAAAUGGUUCA[dT][dT] 3′ and were used. The final amount of siRNA was 180 pmol.

### Stimulation of HEK293 cells

Ionophore stimulation of HEK293 cells was performed to increase intracellular calcium concentration. Ionomycin calcium salt from Streptomyces conglobates (Sigma Aldrich, #I0634) was solved in 50% ethanol and stored in aliquots at −20 °C. Ionophore (final concentration between 1 and 5 μM) and $CaCl_2$ (final concentration 4 mM) were diluted in prewarmed HBSS (Gibco, #14025-092). Cells were washed once with prewarmed DPBS (Gibco, #21969-035) and treated with stimulation solution for 10 min at 37 °C, 5% $CO_2$. Unstimulated controls were treated only with high $CaCl_2$ concentrations. Cells were harvested and proteins were isolated. For 2D gel electrophoresis 70 μg total protein was directly precipitated by TCA to avoid native dephosphorylation.

### Co-immunoprecipitation

For myc tag specific co-immunoprecipitation[62] HEK293 cells were lysed in lysis buffer (in mM): 10 Tris-HCl, pH 7.5, 150 NaCl, 0.5 EDTA, 0.5% NP-40, 1 PMSF, protease inhibitor (complete tablet, Roche #04693116001) and phosphatase inhibitor (EasyPACK tablet, Roche #4906845001). 10 μg of total protein extract was used as input on SDS-Page. 20 μL Myc-Trap magnetic beads (ytma-20, Chromotek) were equilibrated in washing buffer (in mM): 10 Tris-HCl, pH 7.5, 150 NaCl, 0.5 EDTA, 0.1% NP-40. 1 mg of protein lysate per sample was incubated with beads over night at 4 °C. Beads were washed and eluted in 30 μL elution buffer (in mM): 200 glycine, pH 2.5. Before SDS-Page followed by immunoblot analysis eluates were neutralized (in mM): 1000 Tris-HCl, pH 8.

### Ascorbate peroxidase (APEX) catalyzed proximity labeling

ELC was amplified from human left ventricular cDNA[38] and integrated in pcDNA3-APEX2-NES (Addgene, #49386). HEK293 cells were seeded in 150 mm dishes in DMEM media without antibiotics and cultured overnight. For overexpression cells were transfected with lipofectamine (Invitrogen, #13778-150). Empty pcDNA3-APEX2-NES serves as negative control. 24 h after transfection, all samples were treated with tyramide (final concentration: 500 μM, Biotin-Phenol, Sigma, #41994-02.9) for 30 min. Half of the samples were additionally stimulated with 30% $H_2O_2$ (Merck, #107209). Streptavidin pull-down was performed using MyOne C1 beads (Invitrogen, #65001) following manufacture's instruction. In brief, 1 mg of isolated proteins was used for precipitation. Eluates were washed with 6 M Urea before analysis on SDS-PAGE. To check for assay efficiency gels were blotted and stained with streptavidin-horseradish peroxidase (Life Technologies, #S-911). LC-MS analysis of one representative experiment was performed as described after trypsin digestion. One LC-MS run was performed for each sample (APEX-ELC with and without $H_2O_2$, APEX-Ctrl with and without $H_2O_2$). After peptide identification Scaffold (version Scaffold_4.8.9, Proteome Software Inc., Portland, OR) was used to visualize MS/MS based peptide and protein identifications. The identified exclusive peptide counts were summarized for each protein. The inclusion criteria were as follows: peptide thresholds: 90% minimum, 2 peptides minimum, peptide FDR: 1.7%; protein FDR: 0%. Specific interaction partners were identified by calculating the foldchange of

unique peptide counts of biotin-labeled proteins in the proximity of ELC (APEX-ELC with $H_2O_2$ sample) and unique peptide counts of the same protein in the APEX-Ctrl sample.

### SDS-page

Protein samples were mixed with Laemmli sample buffer (Biorad, #1610747) and denatured for 5 min at 95 °C. Protein samples were run on hand-cast polyacrylamide gels (12% separating gel, 5% stacking gel, Mini-PROTEAN® Cell Biorad) in 1× running buffer (in mM): 25 Tris, 192 glycine, 0.1% SDS. Gel run was performed for approximately 1.5 h at 20 mA per gel.

### Immunoblot

Proteins separated by 2D gel electrophoresis or SDS-Page were transferred to a PVDF membrane (Mini Trans-Blot® Electrophoretic Transfer System) for 2 h at 400 mA in 1× blotting buffer (in mM): 19 Tris, 144 glycine, pH 8.3, 15% methanol. 5% milk powder in TBST buffer was used to block unspecific binding. After staining with primary antibody and secondary HRP-antibody, chemiluminescent substrate (ECL Western Blotting Substrate Pierce™ #32106) was used to detect HRP activity from antibodies. Densitometric analysis was performed with ImageJ version 1.51.

### Reblotting of PVDF membranes

PVDF membranes were incubated for 24 min at 50 °C in a harsh stripping buffer: 62 mM Tris-HCl pH 6.8, 2% SDS, 0.8% beta-mercaptoethanol. Membranes were washed intensively with tap water and incubated twice with TBST buffer for 5 min. After additional blocking with 5% milk powder in TBST buffer for at least 1 h, membranes were stained with primary antibodies.

### Zebrafish strains

Care and breeding of zebrafish (Danio rerio) experiments were performed under institutional approval of the Regierungspräsidium Karlsruhe (T52/17), which conform to the Guide for the Care and Use of Laboratory Animals published by The US National Institute of Health (NIH Publication No. 85-23, revised 1996). The AB zebrafish line (European Zebrafish Research Center, ZDB-GENO-960809-7, https://zfin.org/ZDB-GENO-960809-7) and the transgenic zebrafish line *lazy susan* (*laz*[m647])[21] (The Zebrafish Information Network ID, ZDB-ALT-980203-503, https://zfin.org/ZDB-ALT-980203-503#summary) were used. For breeding of zebrafish, piscine eggs were collected, pooled, rinsed, and placed in an incubator at 28 °C until 7 days post fertilization (dpf) upon which they were transferred into the fish room system. Maintenance took place at constant temperature of water and the surrounding at 28 °C and a defined light/dark cycle of 14/10 h. The aquarium system was manufactured by Schwarz (Göttingen, Germany) and enabled constant circulation of water for oxygen supply. Mature fish were fed twice a day with a combination of freshwater aquarium flake food (TetraWerke, Melle, Germany) and live artemia shrimps (Sanders, Great Salt Lake, Artemia cysts).

### Fluorescence in situ hybridization

In-situ probes against zebrafish *nek9* or *camk2g1* were generated by specific amplification using Phusion DNA Polymerase (Thermo Scientific, #M0503). The primer sequences were as follows: *nek9*-F: 5′ ATGTCTCTGGAGGAGTACG 3′, *nek9*-R: 5′ CACACTGCT-GATGTTGGTC 3′, *camk2g1*-F: 5′ GACATCGTCGCCAGAGAGTA 3′, *camk2g1*-R: 5′ GCTCTGCGGTGATTCTCTTG 3′. PCR amplicons were cloned into the pCR® II-TOPO vector (Topo-TA-Cloning Kit, Invitrogen, #K465001) with dual promoter sites according to the manufacturer's instructions. Plasmid sequences were checked by Sanger sequencing before use. After linearization of the plasmids, digoxigenin-(DIG) labeled RNA probes and antisense control

probes were synthesized by using the DIG-RNA Labeling Kit (Roche, #11277073910). Fluorescence in-situ hybridization was performed as described by Brend et Holley[63] with minor modifications. In brief, zebrafish embryos were fixed with 4% paraformaldehyde (Electron Microscope Sciences, #15710) at 72 hpf and dehydrated with 100% methanol. Permeabilization of embryos was performed with proteinase K digestion (5 µg/mL, Invitrogen, #100005393) for 35 min at room temperature. To block unspecific binding prehybridization is performed with heparin and 5 µg/mL torula (yeast) RNA. Hybridization of DIG-labeled probes was carried out at 65 °C overnight. After washing embryos were incubated with anti-Digoxigenin-POD antibody (1:1000, Roche, #11207733910) overnight at 4 °C. For fluorescence labeling and signal amplification the Tyramide Signal Amplification (TSA) plus Fluorescein System (PerkinElmer, NEL741001KT) was used. Imaging was obtained in 3% methylcellulose using a Leica MZ FLIII confocal microscope.

### Antisense-mediated knockdown and sensitizing in zebrafish

Morpholinos-modified antisense oligonucleotide were obtained from Gene Tools (Philomath, OR) and injected into one cell stage zebrafish embryos. Sequences are as follows:

MO-5bp-mm-control-*nek9*: 5′ CTCCTAGTCGTCCAGACACATCATC 3′;
MO-5′UTR-*nek9*: 5′ TCGAGTCTCCACACTGAACTACACA 3′;
MO-*nek9*: 5′ CTCGTACTCCTCCAGAGACATGATC 3′;
MO-*ilk*[26]: 5′ GGCACTGAGTGAAGATGTCATCCAT 3′;
MO-*control*: 5′ CCTCTTACCTCAGTTACAATTTATA 3′.

An injection ramp made of 3% agarose was produced with an injection mold TU-1 (Adaptive Science Tools, Worcester, USA), which contained asymmetric gullies (5 cm length × 3 mm width). The injection needles were made of 1 mm glass capillaries (World Precision Instruments, Sarasota, USA) with a Narishige PC-10 device (Tokyo, Japan) with two heating steps of 72 and 67 °C. The embryos were placed into the asymmetric gullies and were injected with the injection device Femtojet® (Eppendorf, Hamburg, Germany) at a capillary pressure of 15 hPa and an injection duration of 0.1 s. The injection pressure was adjusted to the injection needle. After microinjection, the embryos were incubated at 28 °C for six hours. Living embryos were transferred into fresh petri dishes. At 20 to 22 h post fertilization (hpf) 0.2 mM 1-phenyl-2-thiourea (PTU) was added to suppress pigmentogenesis.

### Rescue experiments in zebrafish

Sense-capped RNA of zebrafish *nek9* and *camk2g1* was generated by using the mMESSAGE mMACHINE™ SP6 Transcription Kit (Thermo Fisher, AM1340). RNA quality and concentration were determined by Fragment Analyzer (Agilent). For rescue experiments the mRNA concentration was titrated in regard to toxicity. Injection needles were calibrated to 1 nL for each injection by using an optical micrometer. Successful injection of mRNA into one-cell stage embryos was followed by Morpholino (MO) injection. Embryos were incubated at 28 °C and analyzed at 72 hpf. Functional analyses of each injected zebrafish clutch were performed blinded to the MO and mRNA injected.

### Assessment of cardiac function

Cardiac function in zebrafish embryos was assessed at 24 or 72 hpf. The embryos were anesthetized with 0.02% tricaine (Sigma-Aldrich, #E10521) and mounted in 2.5% methylcellulose. Video documentation was performed by recording 10 sec video frames using the LEICA DM IRB microscope with a magnification of 1.6×, a LEICA 506059 objective and a LEICA DFC360 FX camera (MCU II, Kappa GmbH, Germany). Images were processed with VirtualDub Version 1.7. Fractional shortening of heart chambers was calculated with help of the zebraFS software (www.benegfx.de).

### CRISPR/Cas9 mediated knockout of *nek9* in zebrafish

CRISPR/Cas9 mediated knockout of zebrafish *nek9* was performed as described by Sorlien et al.[64]. In brief, specific guided RNAs (gRNA) targeting *nek9* were designed. Sequences are as follows:

*nek9* gRNA1: 5′ AACTGTTCACCACGATACT 3′; *nek9* gRNA2: 5′ GGCCGGTCTGCAGCGTCCGG 3′; *nek9* gRNA3: 5′ GGGCAAGGGCGCGTTCGGAG 3′; *tyr* gRNA: 5′ GGACTGGAGGACTTCTGGGG 3′. Full length scaffold gRNAs were synthesized by using the in-vitro transcription kit (New England BioLabs #E3322). Cas9 (PNA Bio #CP01) and gRNAs were combined in a ratio of 2:1 to a final concentration of 400 pg/nL Cas9 protein and 200 pg/nL gRNA. Injection was performed in zebrafish one-cell stage. Additional injection of gRNA against tyrosinase generates mosaic pigmentation phenotypes[65]. For the analysis of efficiency and indel formation the heteroduplex motility assay was used. Two flanking primer pairs were designed: HMA-*nek9*-F1: 5′ GTTCAGTGTGGAGACTCGAGA 3′, HMA-*nek9*-R1: 5′ CCGTCAGGTAAACAGTGCTG 3′, HMA-*nek9*-F2: 5′ ATGTCTCTGGAGGAGTACGAGC 3′, HMA-*nek9*-R2: 5′ AATGTAACGATGTGCAGTGGTC 3′. After PCR amplification samples were denatured for 3 min and cooled down to room temperature for about 1 h. Heteroduplex products were separated by polyacrylamide gel electrophoresis.

### Genotyping of transgenic *nek9* zebrafish embryos

DNA was isolated from fin biopsy of 3-month-old adult zebrafish or from whole embryo at 24 hpf or 72 hpf. PCR amplifications with specific primer pairs flanking the gRNA binding site were performed. F1: 5′ ATGTCTCTGGAGGAGTACGAGC 3′; R1: 5′ ATCTCCTTCCACACCACCAG 3′; F2: 5′ CACTACGCCTCCCTGAACTC 3′; R2: 5′ GCTGATGTTGGTCTGGTTCC 3′.

### Statistics and reproducibility

If not stated otherwise, the results are expressed as mean ± standard deviation (SD). The sample size is listed as n. For zebrafish experiments the total number of analyzed individuals (biological replicates) and the number of independent experiments (technical replicates) is stated. Information about statistical analysis is provided in the figure legends. $P$-values < 0.05 were considered statistically significant. *$P < 0.05$; **$P < 0.01$; ***$P < 0.001$ and ****$P < 0.0001$ are shown in figures. The exact $p$-values are provided in the Source Data file. All statistical analysis was performed using GraphPad Prism Version 5.0 for Windows.

### Reporting summary

Further information on research design is available in the Nature Research Reporting Summary linked to this article.

## Data availability

The authors declare that the data supporting the findings of this study are available within the paper, its Supplementary Information files and Source Data file. The mass spectrometry proteomics data have been deposited to the ProteomeXchange Consortium via the PRIDE[66] partner repository with the dataset identifier PXD016357, PXD034115 and PXD033908. The dataset with the project accession number PXD016357 contains the data presented in Fig. 1B–D and Supplementary Fig. 1. The dataset with the accession number PXD034115 belongs to the APEX interaction screen presented in Fig. 1F. The dataset with the accession number PXD033908 contains the analysis of the myosin light chain migration pattern in 2D gels of human left ventricular proteins presented in Figs. 7A–C and 8E, F. Corresponding run and sample IDs are shown in Supplementary Fig. 10. Phosphorylated peptides assigned to ELC are manually verified and representative spectra are presented in Supplementary Fig. 7. The mRNA sequencing data shown in Fig. 8A are available at https://ccb-web.cs.uni-saarland.de/cms[38]. Source data are provided with this paper.

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

## Acknowledgements

We thank A. Weber, C. Scheiner, and R. Nietsch for great technical support. We thank the Core Facility for Mass Spectrometry & Proteomics at the ZMBH, University Heidelberg, especially S. Merker. We thank W. Rottbauer for his mentoring and co-establishment of the *lazy susan* model. E.K. was partially funded by the Olympia Morata-Program of medical university Heidelberg. N.F. received funding from the Deutsche Forschungsgemeinschaft (DFG FR1289/13-1, DFG-1550-B03, DFG-1550-B10) and the German Centre for Cardiovascular Research (DZHK). B.M. received funding from the Deutsche Forschungsgemeinschaft (DFG ME 3859/4-1 and CRC 1550), the German Centre for Cardiovascular Research (DZHK), Informatics for Life (Klaus Tschira Foundation), the Leducq Foundation (CASTT)I, the Ministry for Science and Education Baden-Wuerttemberg (Precision Digital Health) and the Else-Kröner Exzellenzstipendium.

## Author contributions

B.M. and M.M. designed the study; M.M., R.E., M.K.L., O.F.T., D.M.B., J.M.K., A.P., S.H., S.S., and C.B. performed experiments; B.M., J.N.B., J.H., E.K., F.S., T.R., R.T., N.G., and M.C. analyzed and interpreted data; J.N.B., K.S.F., J.H., A.Ke., and A.Kl. provided research support and conceptional advice; M.M. and B.M. wrote the paper; H.K., H.A.K., N.F., and L.M.S. revised the paper.

## Funding

## Competing interests

The authors declare no competing interests.

## Additional information

[1]Kardiogenetikzentrum Heidelberg, University Hospital of Heidelberg, Heidelberg, Germany. [2]DZHK (German Centre for Cardiovascular Research), Partner Site Heidelberg/Mannheim, Berlin, Germany. [3]Clinic for General and Interventional Cardiology/ Angiology, Herz- und Diabeteszentrum NRW, University Hospital of the Ruhr-Universität Bochum, Bad Oeynhausen, Germany. [4]Clinic and Polyclinic for Cardiology, University of Leipzig, Leipzig, Germany. [5]Department of Medicine III, University of Heidelberg, Heidelberg, Germany. [6]CFMP, Core Facility for Mass Spectrometry & Proteomics at ZMBH, Heidelberg University, Heidelberg, Germany. [7]ZMBH, Center for Molecular Biology, Heidelberg University, Heidelberg, Germany. [8]Clinical Bioinformatics, Saarland University, Saarbrücken, Germany. [9]Department of Neurology and Neurological Sciences, Stanford University Medical School, Stanford, CA, USA. [10]Stanford Genome Technology Center, Stanford University Medical School, Stanford, CA, USA. ✉e-mail: benjamin.meder@med.uni-heidelberg.de

