## [Peer Review File · Nature Communications]

Reviewers' Comments:

Reviewer #1:

Remarks to the Author:

The authors of this study report two kinases (NEK9 and CamK2G) as potential regulators of essential MLC phosphorylation and control of cardiac contractility, with upregulation of p-ELC in patients with distal cardiomyopathy. My review will focus on the initial findings and results relevant to the protein identification and detection.

My first major concern relates to the immunoblotting data. Human ELC is detected using an antibody from Biozol and zebrafish ELC detected using a 'custom' antibody but no further information is provided (eg product codes, epitopes etc). No validation of the antibodies is provided – do they cross react with RLC, for example? How linear is their detection, and is the quantitation by densitometry based on detection within the linear range? More reliable quantitation is achieved for example using fluorescent secondary antibodies rather than ECL. The migration pattern by 2-DE is consistent with mono and bi-phosphorylated products, but can this be confirmed using mass spectrometry?

My second major concern relates to reporting and lack of detail concerning the methods for protein identification. There is only one reference to mass spectrometry (Methods, line 356), however no detail regarding the data acquisition parameters, or liquid chromatography instrumentation or method is provided. Neither are any of the criteria for protein identification (MS1 and MS2 ion tolerances, peptide modifications, number of missed trypsin cleavages etc) provided. Raw MS data should be made available via uploading to a public repository such as ProteomExchange.

Reviewer #2:

Remarks to the Author:

In this manuscript, the authors identified two ELC interacting proteins, NEK2 and CamK2G, and provided evidences to suggest that they might regulate phosphorylation of ELC. The manuscript started with characterization of increased ELC phosphorylation in human LV tissues of DCM patients. They then identified two proteins from coIP that was followed by mass spectrometry. Both in vitro and in vivo experiments were conducted to show that the interaction of ELC with these two kinases is dependent on Ca⁺⁺, KO of either genes in zebrafish affect cardiac function, and both proteins mediate ELC phosphorylation. There are certain novel discoveries made by the authors; however, the experiments were not rigorously conducted, and evidence is not sufficient to support their conclusion, especially in vivo experiments.

Major concerns.

Fig. 4. Specificity of MOs needs to be more carefully examined using more than one approach, including 1) test more than one MO for each gene; 2) conduct rescue experiments by co-injection of mRNA, 3) whenever possible, stable KO lines need to be checked.

The embryonic zebrafish model is an oversimplified model for studying cardiac contractility. FS was used as the sole index, which is not sufficient. Many genes might affect FS in this system.

Phenotypic characterization of morphants is sketchy with insufficient details.

Minor concerns

The nomenclature of Laz mutants is messy. First, all mutants shall be on italics. Second, it can generate confusion if Laz^{+/+} refers to homozygous mutants. Please follow the guidelines recommended by the zebrafish nomenclature committee on zfin.org.

Line 243-246: is an overstatement.

Line 261-262, and Line 298: What is the rationale to hypothesize an adaptive mechanism?

Expression of the two genes during zebrafish embryogenesis needs to be characterized. Whole mount in situ is convenient to show how specific their cardiac expression could be.

It is interesting to note that *camk2g1* morphants manifest ventricular phenotypes only. Why atrium is not affected?

Reviewer #3:

Remarks to the Author:

The manuscript, "Kinases NEK9 and CamK2G regulate phosphorylation of the essential myosin light chain in the heart" by Müller et al., aims to elucidate the role of myosin essential light chain phosphorylation for heart contractility, and identify, for the first time, the associated kinase(s). Protein phosphorylation was analyzed by performing 2D-SDS-PAGE on left ventricular samples from DCM patients versus controls, constituting LV samples from transplanted human hearts (HTX).

It has been well established that human ventricular myosin can be phosphorylated at Ser15 on the regulatory light chain (RLC), and that the kinase responsible for RLC phosphorylation is myosin light chain-specific kinase (MYLK3 gene). It is also known that RLC phosphorylation plays important roles in heart contractility of normal healthy hearts and in heart disease where it was shown to play a role in alleviating the consequences of disease-causing mutations in myosin RLC. In contrast, very little is known about the phosphorylation of myosin essential light chain and its potential role in cardiac muscle physiology. The phosphorylating kinase(s) are also unknown. In this respect, the manuscript by Muller et al. is significant as it addresses an important problem in cardiac muscle physiology associated with the significance of sarcomeric protein phosphorylation for heart contractility, and especially a largely unknown phenomenon, that is, myosin ELC phosphorylation.

More than 15 years ago, using pharmacologically preconditioned cardiomyocytes, the Van Eyk's group identified Ser195 on myosin ELC as a phosphorylatable residue. Proteomic analysis also identified a second potential site of ELC phosphorylation at Thr69. The physiological significance of Ser195-ELC phosphorylation was later demonstrated in zebrafish system by Meder et al. 2009, and by Scheid et al. Using C-terminally truncated mutant (*lazm647*) of ELC, severe contractile insufficiency and early embryonic death were observed. Similar ELC-mutant associated defects in heart contractility were also monitored in adult zebrafish, suggesting that Ser195 phosphorylation of ELC plays a critical role for actin-myosin interaction and heart performance. The innovation of the current study is the identification of ELC interacting kinases: NIMA related kinase 9 (NEK9), and calcium/calmodulin dependent kinase II gamma (CamK2G), both expressed in the heart, and observed to interact with myosin ELC in a calcium-dependent manner.

To this reviewer, there are some problems with the data presented in the manuscript. First, there is lack of evidence supporting the observation that two of the three detected ELC protein spots in 2D-SDS-PAGE on left ventricular samples from DCM vs transplanted heart controls (HTX) in Fig. 1A, represent the mono- (1P) and dual- (2P) phosphorylated ELC. There is no proof that the band labeled as 1P is not an atrial ELC isoform (ELCa). It has been well established that ELCa is expressed in the ventricles of human hearts, and that its expression is significantly augmented in cardiac disease. The isoelectric point (pI) for the ventricular ELCv isoform is pI=5.03, MW 21,932, while that of atrial ELCa is pI=4.98, MW 21,550. It is not surprising that transplanted hearts would express ELCa, and that DCM samples showed significantly increased ELCa. In fact, increased ELCa expression is often considered as a marker of heart dysfunction in DCM and heart failure.

Likewise, the protein spots for the myosin regulatory light chain (RLC), labeled as 1P or 2P, have not been confirmed, either by mass spec analysis or by using phospho-specific RLC antibodies. It is not clear why there are two RLC phosphorylation variants as it is well established that the human ventricular RLC has one functional phosphorylation site at Ser15. The smaller acidic spots in Figs. 1a and 1b do not necessarily denote a double phosphorylated RLC, and they could represent

deamidated RLC at Asp14 variant (Scruggs et al. 2012) with pI=4.770. Deamidation of RLC would result in a phosphorylation-like leftward shift on the horizontal axis of 2D SDS-PAGE, where proteins are separated by isoelectric point.

In summary, the 2D immunoblot data (Figs. 1 and 6) that the authors present as evidence for the existence of phosphorylated forms of ELC and RLC proteins in the left ventricles of DCM/HTX patients only show their charged variants, and a solid evidence that they indeed represent the phosphorylated forms of ELC/RLC is missing. This is especially important for myosin ELC, whose phosphorylation status and association with cardiac disease is not well understood. The fact that there is an interaction between the ELC and CamK2G (Fig. 3) does not necessarily prove that the ventricular ELC protein becomes phosphorylated in LV samples of DCM vs HTX hearts. The authors are encouraged to provide convincing evidence (by mass spec or using ELC-specific phospho-antibodies) that the observed charged variants detected in the LV samples are indeed the phosphorylated forms of myosin essential light chains.

Minor:

1. Introduction: The statement: "For example, phosphorylation of the RLC protein at amino acid position serine 15 by the myosin light chain kinase (MLCK) is pivotal for the contraction of smooth muscle cells" is not quite right as the smooth muscle myosin is phosphorylated at Ser19/Thr18.

2. Isoelectric point should read isoelectric.

Reviewer #4:

Remarks to the Author:

The authors showed upregulated ELC phosphorylation levels in normal and failing hearts of DCM patients. They identified NEK9 and CamK2G as ELC interaction partners. Silencing of CamK2G and/or NEK9 results in blunting of ELC phosphorylation in vitro. Loss of kinase function leads to severe impairment of cardiac contractility in vivo.

The paper is well-written but there are some major problems.

Main problems:

The key results of the presented paper are based on 2D-PAGE analysis of myosin light chains, which seems to be misinterpreted in many aspects.

Fig. 1a shows 2D-PAGE analysis of ELC and RLC of control and DCM human tissue. Albeit the RLC was separated into four distinct protein spots, the authors interpreted these protein spots as non-P, mono-P, and di-P - without any experimental evidence. Furthermore, this interpretation is in disagreement with data from the literature: it could already be shown that human ventricular RLC is mono-P only, the other spots in the 2D-PAGE are derived from post-transcriptional modification by deamination causing a similar shift of the pI like phosphorylation.

The authors should correct their manuscript accordingly. In addition, Scrugges et al., 2010 already compared normal with DCM RLC forms, which should be noted in a revised manuscript.

The authors interpreted their 2D-PAGE of the ELC in a similar way as described above for the RLC. They named the ELC spots non-P, mono-P, and di-P, again without any evidence. Alternative post-transcriptional modifications of the ELC which could mimic phosphorylation in the 2D-PAGE were not considered at all.

Therefore, it is obligatory that the authors experimentally show that 1P and 2P of human ventricular ELC are indeed mono- and di-phosphorylated forms and determine the amino acid residues which are assumed to be phosphorylated. This seems to be important, since ELC showed multiple phosphorylation sites besides T64 und S195 (Arrell et al., 2001) (Cadete et al., 2012). The complex phosphorylation pattern of cardiac ELC was even raised upon ischemia-reperfusion

experiments of rat hearts (Cadete et al., 2012).

Minor problems:

Lines 71-73, Ref. 11-16: the References which described it first should be cited

Line 261, Ref. 16: no ELC phosphorylation measured in this reference

Reviewer #1:

My first major concern relates to the immunoblotting data. Human ELC is detected using an antibody from Biozol and zebrafish ELC detected using a 'custom' antibody but no further information is provided (eg product codes, epitopes etc).

We added the information in the Materials and Methods section. The Biozol ELC antibody (Catalog number: ZF127578) is polyclonal and the company does not provide epitopes. The antibody against zebrafish ELC was ordered as a service for custom polyclonal antibody production by Eurogentec using immunization of rabbits with the peptide NH₂-CAPVPETPKEPEVDLK-CONH₂. Quality control of the purified antibody was performed by ELISA and SDS PAGE by Eurogentec and validation by us has been performed in this revision (see below).

No validation of the antibodies is provided – do they cross react with RLC, for example? How linear is their detection, and is the quantitation by densitometry based on detection within the linear range? More reliable quantitation is achieved for example using fluorescent secondary antibodies rather than ECL.

We thank the reviewer for this important point. We have now done multiple experiments to further validate the used antibodies and ECL detection method. As shown in Suppl. Fig. 2, the Biozol ELC antibody does only detect ELC (22 kDa) but does not significantly cross-react with RLC, which is predicted to result in a band at 19 kDa. Using a titration of protein input amounts followed by IB and different exposure times, we also show that linear quantification can be done on immunoblotting in the input protein range from 20-50 µg, which was used as input throughout the experiments in this study. For comparability of the many different experiments based on our established assay (in total 54 MLC immunoblots have been performed (Fig. 1d, Fig. 6a,b,c, Suppl. Fig. 7c,d) and the method was also used in previously published articles e.g. in Scheid *et al.*, *Cardiovasc Res.* 2016 Jul 1;111(1):44-55), we did not move to a new detection system, however, we agree that this can be implemented for further experiments.

The migration pattern by 2-DE is consistent with mono and bi-phosphorylated products, but can this be confirmed using mass spectrometry?

We thank the reviewer for this comment. To be more precise we renamed the ELC 2D migration pattern. The most basic spot is indicated as "Basal". Two additional spots shifted towards an acidic pI can also be seen. These spots were labeled "+1P" and "+2P" suggesting the introduction of one or more phosphorylations at residues we now identified via mass spec (see Fig. 1b).

In order to provide evidence on the nature of the posttranslational modifications, we performed dephosphorylation experiments in human tissue as suggested by the reviewers. Protein isolates from human tissue (DCM and control) were treated with calf intestinal alkaline phosphatase (CIP) followed by 2D with IB. As shown in Fig.1c, the predicted phosphorylation forms with acidic pI have almost completely disappeared. Only a very weak spot shifted towards acidic pI is remaining. For

RLC, the results are more indicative of deamidation as suggested by one reviewer, since the acidic shifted form is not disappearing after phosphatase treatment (Suppl. Fig. 1).

My second major concern relates to reporting and lack of detail concerning the methods for protein identification. There is only one reference to mass spectrometry (Methods, line 356), however no detail regarding the data acquisition parameters, or liquid chromatography instrumentation or method is provided. Neither are any of the criteria for protein identification (MS1 and MS2 ion tolerances, peptide modifications, number of missed trypsin cleavages etc) provided.

We apologize for the missing information, which we now provide in the Materials and Methods section paragraph "Liquid chromatography mass spectrometry". We also included all experimental setup conditions for the new mass-spec experiments.

Raw MS data should be made available via uploading to a public repository such as ProteomExchange.

The mass spectrometry data have been deposited to the ProteomeXchange Consortium via the PRIDE partner repository with the dataset identifier PXD016357 and PXD016359.

Project Name: Identification of ELC interacting proteins in zebrafish heart. **Project accession:** PXD016357 (**Username:** reviewer60495@ebi.ac.uk, **Password:** XJixTIEB)

Project Name: Analysis of myosin light chain migration pattern in 2D gels of human left ventricular proteins. **Project accession:** PXD016359 (**Username:** reviewer88570@ebi.ac.uk, **Password:** z6LNfwvg)

Reviewer #2:

Specificity of MOs needs to be more carefully examined using more than one approach, including:

1) test more than one MO for each gene

We thank the reviewer for the important notes and suggestions for improvements. We have now performed additional experiments using alternative Morpholino antisense oligonucleotides. For *nek9* experiments, we designed a 5-prime UTR targeting MO resembling the cardiac phenotype, which we describe now more thoroughly in the results section and Suppl. Fig. 5. For the controls, we used 5-mismatch MO for *nek9* and *camk2g1* and integrated the data in Suppl. Fig. 4.

2) conduct rescue experiments by co-injection of mRNA,

We performed the requested rescue experiments with the zebrafish mRNA of *nek9* and *camk2g1* and analyzed all embryos not only qualitatively but also quantitatively. The analysis was blinded to

the MO that was injected and groups were only retrospectively annotated. As shown in Fig. 4 c-h, the cardiac phenotype after knockdown of *nek9* and *camk2g1* can be mostly rescued, qualitatively and quantitatively. In detail, when injecting 150 ng mRNA (*in vitro* transcription) followed by 45 μ M MO-*nek9* in one-cell stage embryos, fractional shortening remains comparable to control injected embryos after 72 hpf. The micro-injection needles were always calibrated to 1 nL using an optical micrometer. The very good Morpholino guide provided by Chien's lab published on zfin.org was carefully followed (<https://wiki.zfin.org/display/prot/Morpholino+tips%2C+Chien+lab>).

3) whenever possible, stable KO lines need to be checked.

To our knowledge no stable genetic KO lines are available for *nek9* or *camk2g1* and the generation by CRISPR-Cas9 with necessary outcrossing over 3-4 generations would take considerable time. We pointed out in the discussion section the potential limitation of anti-sense experiments in cell cultures and zebrafish.

The embryonic zebrafish model is an oversimplified model for studying cardiac contractility. FS was used as the sole index, which is not sufficient. Many genes might affect FS in this system. Phenotypic characterization of morphants is sketchy with insufficient details.

We appreciate the comment and now extended our phenotype description in the Results section "NEK9 and Camk2G are essential for cardiac function in zebrafish". We also enhanced the cardiac phenotype description and performed quantitative analysis of heart rate. We also improved description of the genetic sensitizing experiment as outlined in Fig. 5a. As shown, low-dose MO-*nek9* (at a dose below resulting in a phenotype), can be sensitized by a heterozygous ELC mutation, which *per se* does not lead to any obvious phenotype in zebrafish embryos. This approach can provoke the *lazy susan* (*laz*^{m647}) phenotype in heterozygous embryos (*laz*^{m647/+}) (genotyping was individually performed after blinded phenotyping and qualitative and quantitative analysis). This is not the case by a MO against e.g. integrin-linked kinase, which targets another pathway controlling cardiac contractility (Bending *et al.*; *Genes Dev.* 2006 Sep 1; 20(17)). This sensitizing experiment suggests a common pathway of Nek9-kinase, ELC-function and contractility.

Minor concerns:

The nomenclature of Laz mutants is messy. First, all mutants shall be on italics. Second, it can generate confusion if Laz+/+ refers to homozygous mutants. Please follow the guidelines recommended by the zebrafish nomenclature committee on zfin.org.

We apologize for this unnecessary confusion. We corrected throughout the manuscript.

Line 243-246: is an overstatement.

We corrected this overstatement.

Line 261-262, and Line 298: What is the rationale to hypothesize an adaptive mechanism?

There are many different causes of heart failure and in many cases the precise cause remains elusive. In this context, we do not believe that altered phosphorylation of ELC is the cause of the disease in the patient samples we studied, but rather a consequence of e.g. increased wall-stress. We tuned the discussion to reflect this hypothesis more clearly.

Expression of the two genes during zebrafish embryogenesis needs to be characterized. Whole mount in situ is convenient to show how specific their cardiac expression could be.

As written in our manuscript and shown by our developmental gene expression analysis (Fig. 2g), we do not postulate a specific cardiac expression. Considering that about 97% of phospho-sites do not have an annotated kinase and gene expression of annotated or proposed kinases are mostly not tissue specific, proposed mechanisms include diverse functions depending on the developmental context, co-factors, stimulators (in our case calcium influx) and others. We did follow the advice of this reviewer and now performed fluorescence in-situ hybridizations. As shown in Suppl. Fig. 4e, *nek9* and *camk2g1* are expressed ubiquitously. In detail, *nek9* and *camk2g1* were localized in zebrafish brain, somites, differentiating muscle and cardiac tissue at 72 hpf. There is a stronger cardiac expression of *nek9* compared to *camk2g1* (shown at 72 hpf).

It is interesting to note that camk2g1 morphants manifest ventricular phenotypes only. Why atrium is not affected?

This is an interesting question, but we can only speculate on this. When we compare the gene expression of *camk2g1* of atrial and ventricular tissue of humans (GTEx query as of Nov. 2019), we see equal expression. It, however, might involve additional co-factors that are not equally expressed or be different in zebrafish and humans. We hinted at this observation in the phenotype description in the results section.

Reviewer #3:

First, there is lack of evidence supporting the observation that two of the three detected ELC protein spots in 2D-SDS-PAGE on left ventricular samples from DCM vs transplanted heart controls (HTX) in Fig. 1A, represent the mono- (1P) and dual- (2P) phosphorylated ELC. There is no proof that the band labeled as 1P is not an atrial ELC isoform (ELCa). It has been well established that ELCa is expressed in the ventricles of human hearts, and that its expression is significantly augmented in cardiac disease. The isoelectric point (pI) for the ventricular ELCv isoform is pI=5.03, MW 21,932, while that of atrial ELCa is pI=4.98, MW 21,550. It is not surprising that transplanted hearts would express ELCa, and that DCM samples showed significantly increased ELCa. In fact, increased ELCa expression is often considered as a marker of heart dysfunction in DCM and heart failure.

We thank the reviewer for his suggestions on further specifying our findings in human tissue. To be more precise we renamed the ELC 2D migration pattern. The most basic spot is indicated as "Basal". Besides the basal ELC spot, two additional spots shifted towards an acid pI can be seen. These spots were labeled "+1P" and "+2P" suggesting the introduction of one or more phosphate moieties. In order to prove the shift towards acidic pI to be caused by posttranslational ELC phosphorylation, we now performed dephosphorylation experiments in human LV tissue from DCM cases and HTX controls. Precisely, we treated protein extracts with calf intestinal alkaline phosphatase (CIP) for 1 hour followed by 2D with IB. As shown in Fig.1c, the predicted phosphorylation forms with acidic pI have almost completely disappeared. Only a very weak spot shifted towards acidic pI is remaining.

To dissect if other, upregulated ELC/RLC isoforms are present, we performed mass-spectrometry analysis after cutting the spot-regions in question from silver-stained 2D gels of protein from the human tissue. As shown in Fig. 1e, we can well distinguish peptides from ELC and RLC with almost full protein coverage. In detail, human left ventricular tissue of four different samples (2 DCM patients and 2 healthy controls) were analyzed by mass spectrometry (Fig 1a). In the spot areas labeled with 1,2 and 3 the ventricular ELC protein (ENSP00000379210 MYL3-002, 22kDa) was detected to be most abundant. In accordance, the spot areas labeled with 4,5 and 6 represent the ventricular RLC with highest abundance (ENSP00000228841 MYL2-001, 19kDa). Only very few peptide counts were detected for other myosin light chain isoforms. Importantly, MYL4 (ELCa) was not detected in DCM at all and showed only 2 unique peptide counts and 1 unique peptide count in controls (compared to 21 unique peptide counts and 18 unique peptide counts for ELCv, respectively).

Likewise, the protein spots for the myosin regulatory light chain (RLC), labeled as 1P or 2P, have not been confirmed, either by mass spec analysis or by using phospho-specific RLC antibodies. It is not clear why there are two RLC phosphorylation variants as it is well established that the human ventricular RLC has one functional phosphorylation site at Ser15. The smaller acidic spots in Figs. 1a and 1b do not necessarily denote a double phosphorylated RLC, and they could represent deamidated RLC at Asp14 variant (Scruggs et al. 2012) with pI=4.770. Deamidation of RLC would result in a phosphorylation-like leftward shift on the horizontal axis of 2D SDS-PAGE, where proteins are separated by isoelectric point.

The reviewer is right and our data underline his suggestions for RLC deamidation (see Suppl. Fig. 1). We provide our new data and the given reference for RLC posttranslational modification.

In summary, the 2D immunoblot data (Figs. 1 and 6) that the authors present as evidence for the existence of phosphorylated forms of ELC and RLC proteins in the left ventricles of DCM/HTX patients only show their charged variants, and a solid evidence that they indeed represent the phosphorylated forms of ELC/RLC is missing. This is especially important for

myosin ELC, whose phosphorylation status and association with cardiac disease is not well understood. The fact that there is an interaction between the ELC and CamK2G (Fig. 3) does not necessarily prove that the ventricular ELC protein becomes phosphorylated in LV samples of DCM vs HTX hearts. The authors are encouraged to provide convincing evidence that the observed charged variants detected in the LV samples are indeed the phosphorylated forms of myosin essential light chains.

As shown in the experiments above, we indeed can prove that the charged variants of ELC are phosphorylated forms by *in vitro* phosphatase treatment. As suggested by the other reviewers, we also consolidated the experimental findings *in vivo*, underlining the functional role of both detected kinases in regulating cardiac contractility and ELC phosphorylation. We also performed genetic sensitizing experiments (overview Fig. 5a). As shown, low-dose MO-*start-nek9*, which is at a dose below resulting in a phenotype, can be sensitized by a heterozygous ELC mutation, which per se does not lead to any obvious phenotype in zebrafish embryos. When applied together, we can provoke *lazy susan* (*laz^{m647}*) phenotype in heterozygous embryos (*laz^{m647/+}*) (genotyping was individually performed after blinded phenotyping qualitatively and quantitatively). This is not the case by a MO against e.g. integrin-linked kinase, which leads to severe heart contractility phenotype at full dose (Bending *et al.*; *Genes Dev.* 2006 Sep 1; 20(17)). When applied in the dose just below the cutoff for phenotype provocation, we do not see a phenotype alone or in combination with one *laz^{m647}* allele.

Minor:

1. Introduction: The statement: "For example, phosphorylation of the RLC protein at amino acid position serine 15 by the myosin light chain kinase (MLCK) is pivotal for the contraction of smooth muscle cells" is not quite right as the smooth muscle myosin is phosphorylated at Ser19/Thr18

Thank you for pointing this out, the reviewer's expertise is really appreciated. We corrected this in the manuscript.

2. Isoelectric point should read isoelectric.

Corrected.

Reviewer #4:

The key results of the presented paper are based on 2D-PAGE analysis of myosin light chains, which seems to be misinterpreted in many aspects. Fig. 1a shows 2D-PAGE analysis of ELC and RLC of control and DCM human tissue. Albeit the RLC was separated into four distinct protein spots, the authors interpreted these protein spots as non-P, mono-P, and di-P - without any experimental evidence. Furthermore, this interpretation is in disagreement with data from the literature: it could already be shown that human

ventricular RLC is mono-P only, the other spots in the 2D-PAGE are derived from post-transcriptional modification by deamination causing a similar shift of the pI like phosphorylation. The authors should correct their manuscript accordingly.

We have performed additional experiment following the very good suggestion of this reviewer. Using both, *in vitro* phosphatase treatment and mass-spectrometry, we can provide evidence for our data given on ELC. First, we performed dephosphorylation experiments treating protein isolates from human tissue (DCM and control) with calf intestinal alkaline phosphatase (CIP) followed by 2D with IB. As shown in Fig.1c, the predicted phosphorylation forms with acidic pI have almost completely disappeared. Only a very weak spot shifted towards acidic pI is remaining. For RLC the results are more indicative of deamidation as suggested by the reviewer, since the acidic shifted form is not disappearing after phosphatase treatment. Next, we performed extensive mass-spectrometry analysis after cutting the protein-spots in question from silver-stained 2D gels of protein from human tissue. As shown in Fig. 1a, we can well distinguish peptides from ELC and RLC with almost full coverage each at the regions stained by IB. Fig. 1b shows the phosphorylation sites detected in our MS experiments. This data shows that more than half of all potential phosphorylation sites are detected to be phosphorylated in human LV tissue (9 out of 17). In contrast, for RLC four distinct phosphorylation sites were identified (4 out of 15). The spectra of all reliable detected phospho-peptides are included in the manuscript (Suppl. Fig. 9). Besides posttranslational phosphorylation of ELC and RLC protein deamidation and oxidation was detected. Deamidation and especially oxidation is known to be potentially introduced non-enzymatically *in vivo* and spontaneously *in vitro*. In addition, artificial deamidation during proteomic digestion for sample preparation may hamper the identification and quantification of deamidation sites. However, we detected deamidated RLC at *Asp14*, as suggested by one reviewer and already published by Scrugges *et al.* 2010. We included the data in our revised manuscript and updated the results for RLC.

Scrugges et al. 2010 already compared normal with DCM RLC forms, which should be noted in a revized manuscript. The authors interpreted their 2D-PAGE of the ELC in a similar way as described above for the RLC. They named the ELC spots non-P, mono-P, and di-P, again without any evidence. Alternative post-transcriptional modifications of the ELC which could mimic phosphorylation in the 2D-PAGE were not considered at all. Therefore, it is obligatory that the authors experimentally show that 1P and 2P of human ventricular ELC are indeed mono- and di-phosphorylated forms and determine the amino acid residues which are assumed to be phosphoylated. This seems to be important, since ELC showed multiple phosphorylation sites besides T64 und S195 (Arrell et al., 2001) (Cadete et al., 2012). The complex phosphorylation pattern of cardiac ELC was even raised upon ischemia-reperfusion experiments of rat hearts (Cadete et al., 2012).

The reviewer again raises an important point. We are happy to provide supporting data on a) the specificity of our antibodies, b) the phosphorylation of ELC by dephosphorylation assay, c) mass-

spectrometry-based identification of the complex ELC-phosphorylation events and d) the genetic sensitizing experiments that underline the joint pathway of NEK9 kinase, ELC phosphorylation and cardiac phenotypes.

Minor

Lines 71-73, Ref. 11-16: the References which described it first should be cited

We apologize.

Line 261, Ref. 16: no ELC phosphorylation measured in this reference

Corrected.

Reviewers' Comments:

Reviewer #1:

Remarks to the Author:

The concerns raised in my review were mostly related to the immunoblotting data. The authors have gone to considerable lengths and provided further data to support their conclusions, and I am satisfied with their responses.

They have also included sufficient information related to the analytical conditions used for the protein identification methods.

As such, and with the availability of the data provided in a public repository, I am satisfied that the proteomic analysis aspect of this paper meet the required standards for publication.

Reviewer #2:

Remarks to the Author:

The authors have answered most of my questions. The manuscript is much improved, with a better focus. I only have a minor concern:

Line 197 and Fig. 4F: Please provide more complete information on the 2 stable mutants for nek9. Are homozygous mutants for 78del and 500 del single mutants embryonic lethal, as suggested by severe cardiac defects in morphants in Fig. 4A? I guess combined heterozygous NEK9 is comparable to a homozygous nek9 mutant, which is also anticipated to be embryonic lethal. However, in Fig. 4F, it appears that this double mutant manifests comparable cardiac defects with the 2 single heterozygous mutants.

Reviewer #3:

Remarks to the Author:

The revised manuscript by Müller et al. provided new evidence for NEK9 and Ca²⁺-dependent phosphorylation of human cardiac myc-ELC in vitro (Figure 3), partially resolving my concerns about ELC and RLC phosphorylation. I commend the authors for performing new experiments with the use of calf intestinal alkaline phosphatase to dephosphorylate both light chains in human cells overexpressing myc-ELC (Figure 3C), and in ventricular tissue from native, DCM, and HTX samples (Figures 5 and 6). They convincingly showed on 2D SDS-PAGE that after the treatment with phosphatase the two charged myc-ELC spots were no longer present when blotted with ELC-specific antibodies (Figure 3C). This result suggested that the charged spots seen on the gel before the treatment with phosphatase were most likely representing mono (+1P) and di-(+2P) phosphorylated ELC. Of note, the blot in Figure 3C could be further confirmed by using myc-specific antibodies.

However, the results of testing native ventricular tissue (Figure 5E, for vELC and Figure 5F, for vRLC) are somewhat confusing. All charged ELC variants disappear after treatment with phosphatase while the charged variants of RLC do not, suggesting that the latter correspond to the deamidated RLC. Respective mass spec data on native ventricular samples show that ELC can be phosphorylated at 9 residues while RLC at four residues including the known RLC phosphorylation site at Ser15. The authors need to reconcile the mass spectrometry data with Western blotting results showing (+1P) and di-(+2P) phosphorylated ELC and (0P)-RLC. They also should propose which of the nine phosphorylated ELC residues form (+1P) and di-(+2P) ELC. The use of mass spec could also be helpful to show the pattern of phosphorylated ELC residues after the samples were treated with phosphatase. Experiments also showed an enhanced ELC phosphorylation in DCM samples compared with controls (Figure 6B), and -as before - the phosphorylated (or charged) ELC variants disappeared on phosphatase treatment. It would be informative to acquire mass spec data on DCM samples ± phosphatase.

Regarding the RLC data, the James Stull group proposed that myosin RLC is constitutively phosphorylated (at Ser15) in the heart at a physiological level of approximately 40% (Chang, Battiprolu et al. 2015). The samples presented as "native" in Figure 5F were incubated with the

phosphatase-specific inhibitor, and it is unlikely that all charged RLC variants in Figure 5F represent deamidated RLC and none the (+1P)-RLC. Western blotting data are not consistent with mass spec results that identified phosphorylated Ser15-RLC and other P-residues of RLC in normal healthy ventricular samples but no deamidated Asn14 was found (Figure 5G). This controversy should be somehow resolved. Unlike for the ELC, there are commercially available phospho-specific RLC antibodies, and the lack of reaction to phosphatase treatment shown in Figure 5F should be demonstrated by Western blotting and phospho-specific RLC antibodies. It would also be interesting to see the RLC phosphorylation status in DCM samples \pm phosphatase.

In summary, the authors are encouraged to provide direct evidence for (+1P)-ELC and (+2P)-ELC and identify the phosphorylated residues. The phosphatase treatment data on charged variants of ELC should be supported by direct evidence showing that 2D SDS-PAGE spots are truly mono- and di-phosphorylated ELC. The phosphatase treatment experiment should include positive controls with the use of other than ELC phosphorylated proteins whose dephosphorylation due to phosphatase treatment could be clearly documented.

With the lack of commercially available phospho-specific ELC antibodies, the phosphorylation of ELC may be confirmed using radioisotope phosphate labeling of protein with radioactive ATP. Other methods could include the Pro-Q Diamond phosphoprotein gel stain that selectively stains phosphoproteins in polyacrylamide gels. Although as stated by the authors "1D-phosphogels were not able to separate different ELC forms from human tissue", they might have tried this method already, but it is not stated in the manuscript. There are numerous ProQ/Sypro images of ventricular tissue from healthy and diseased hearts available in the literature showing a positive reaction to ProQ of phosphorylated forms of MyBP-C, Troponin T, and Troponin I, Tropomyosin, and myosin RLC. This reviewer is not aware of any ProQ/Sypro images that were able to detect phosphorylated ELC. Likewise, there are many published images of phosphorylated RLC using Urea/glycerol gels in which the phosphorylated form of RLC migrates faster than the non-phosphorylated RLC. None of such images exist to demonstrate ELC phosphorylation.

On another issue, it was observed that in response to ionophore stimulation and increased $[Ca^{2+}]$, the phosphorylation of ELC is significantly enhanced and an additional phosphorylated ELC form (+2P) becomes detectable (Figure 3D). The authors do not discuss any potential mechanism for this observation. Does Ca^{2+} bind to NEK9 activating the kinase in a Ca^{2+} concentration-dependent manner?

Danuta Szczesna-Cordary

Reviewer #4:

Remarks to the Author:

Although the authors presented a revised manuscript with new information, many major concerns remain.

The authors could not meet my concerns about identification of ELC phosphorylation. One main experiment of the authors to prove that ELC is mono- and diphosphorylated (1P, 2P, resp.) is dephosphorylation of ELC and RLC with a phosphatase (Figures 3C, 5E, and 5F). It is clear from Fig. 3C and 5F that loading of ELC after phosphatase treatment is much smaller than ELC loading in the native state. The presence of phosphorylated spots, therefore cannot be detected. This could easily explain the lack of 1P and 2P after phosphatase treatment and has nothing to do with ELC dephosphorylation. Furthermore, I doubt that the phosphatase activity was sufficient for light chain dephosphorylation. This can be deduced from Figure 5C, i.e. native and dephosphorylated RLC. Protein loading in this Figure was comparable between native and phosphatase treated RLC, but no effect of phosphatase treatment on RLC phosphorylation could be demonstrated in this experiment. This shows that the phosphatase experiment performed by the authors failed in general. The authors left the referee with the question of the nature not only of the ELC but also of the RLC spots.

It was nice to see that the authors could confirm the presence of multiple (9) ELC phosphorylation sites within the ELC sequence (Fig. 5G). But the referee was rather disappointed that the authors did not even try to identify the phosphorylated amino acid sites supposed to be ELC- 1P and 2P.

Furthermore, in case there are 9 amino acid sites phosphorylated, one would expect to see 10 ELC protein spots in the 2D-PAGE, far above the 1P and 2P presented in the revised manuscript. Likewise, the authors showed three more amino acid sites phosphorylated, besides the known S15 of RLC. If this is true, then the RLC should show 8 protein spots in the 2D-PAGE, far more than the two spots shown in Figure 5F. The authors did not give any reasonable explanation of these obvious contradictions.

There are some more major concerns. The authors claimed to have identified kinases which phosphorylate ELC. But it is not clear at all which ELC amino acid sites were phosphorylated by these kinases. In addition, closely mapped protein-protein interaction sites between ELC and the kinases should be presented. This is basic information required for publication in a major journal. With Figure 3D the authors tried to show that NEK9 activity is calcium dependent: ELC phosphorylation under basal condition rose with increasing calcium, an effect which disappeared in case of NEK9 knock down. However, if NEK9 is overexpressed, ELC phosphorylation rose somewhat even without calcium, while additional treatment with calcium was without any effect on ELC phosphorylation. Again, this contradiction is not explained at all.

Response to the Reviewers:

Reviewer #1:

The authors have gone to considerable lengths and provided further data to support their conclusions, and I am satisfied with their responses. They have also included sufficient information related to the analytical conditions used for the protein identification methods. As such, and with the availability of the data provided in a public repository, I am satisfied that the protein/proteomic analysis aspect of this paper meet the required standards for publication.

We thank the reviewer for her/his comments that helped to further improve our manuscript and for her/his recognition of the quality of our experimental approach.

Reviewer #2:

The authors have answered most of my questions. The manuscript is much improved, with a better focus.

Thank you for the positive perception of our work and your constructive comments, which heavily improved our manuscript.

I only have a minor concern: Are homozygous mutants for 78 del and 500 del single mutants embryonic lethal, as suggested by severe cardiac defects in morphants in Fig. 4A? I guess combined heterozygous NEK9 is comparable to a homozygous nek9 mutant, which is also anticipated to be embryonic lethal.

We added additional information regarding the CRISPR/Cas9 lines lacking NEK9 and performed additional experiments (Figure 4 H, J and Supplemental Figure 4 D). We now present sequencing results (Supplemental Figure 4 D) and overexpression analysis (Figure 4 H, I) showing that the deletion in allele *nek9^{78del}* leads to a lack of the ATP-binding site and parts of the protein kinase domain of NEK9. *nek9^{+/500del}* is harboring a frame shift mutation resulting in a premature stop codon (Supplemental Figure 4 D). Regarding your specific question, in the F1 generation we either identified adult *nek9^{+/+}* wild-types or heterozygous mutant fish supporting your presumption that homozygous mutants are embryonic lethal. In addition, at 72 hpf only 8% of the characterized embryos after incrossing of the two different NEK9 mutant lines (Incross *nek9^{78del/+}* x *nek9^{+/500del}*) were combined heterozygous mutants (*nek9^{78del/500del}*), indicating your point of a more detrimental effect in combined heterozygosity (Supplemental Figure 4 C). We added these points to the revised manuscript.

Line 197 and Fig. 4F: Please provide more complete information on the 2 stable mutants for nek9. However, in Fig. 4F, it appears that this double mutant manifest comparable cardiac defects with the 2 single heterozygous mutants.

The cited analysis of course only includes surviving embryos. Heterozygous *nek9^{+/500del}* as well as combined heterozygous *nek9^{78del/500del}* mutant embryos show a significantly higher penetrance to develop heart failure (Figure 4 E). Since allele *nek9^{500del}* undergoes nonsense-mediated mRNA decay (NMD) and *nek9^{78del}* 'only' has disturbed ELC binding (Figure 4 H, I), there might not be an asymmetric effect of both alleles (loss-of-function effect vs. dominant negative effect). We now present the interaction of ELC with the different mutant protein forms of piscine and human NEK9, which supports

this hypothesis. In summary, each mutant allele adds a detrimental effect on survival and heart failure occurrence.

Reviewer #3:

The revised manuscript by Müller et al. provided new evidence for NEK9 and Ca²⁺-dependent phosphorylation of human cardiac myc-ELC in vitro (Figure 3), partially resolving my concerns about ELC and RLC phosphorylation. I commend the authors for performing new experiments with the use of calf intestinal alkaline phosphatase to dephosphorylate both light chains in human cells overexpressing myc-ELC (Figure 3C), and in ventricular tissue from native, DCM, and HTX samples (Figures 5 and 6). They convincingly showed on 2D SDS-PAGE that after the treatment with phosphatase the two charged myc-ELC spots were no longer present when blotted with ELC-specific antibodies (Figure 3C). This result suggested that the charged spots seen on the gel before the treatment with phosphatase were most likely representing mono (+1P) and di-(+2P) phosphorylated ELC. Of note, the blot in Figure 3C could be further confirmed by using myc-specific antibodies.

Dear Danuta, thank you for your important comments that we carefully considered and that further improved our story. The images in revised Figure 3 C show blots using a myc-specific antibody, detecting only our overexpressed myc-ELC in HEK cells. We believe this is a strong experiment, since we can exclude any intrinsic protein and our exogenous tagged protein completely resembles the phosphorylation pattern of ELC in the myocardium. As shown, overexpression of NEK9, as well as absence of NEK9 have opposing effects. We added this information and clarified the use of myc-tagged ELC forms.

The results of testing native ventricular tissue (Figure 5E, for vELC and Figure 5F, for vRLC) are somewhat confusing. All charged ELC variants disappear after treatment with phosphatase while the charged variants of RLC do not, suggesting that the latter correspond to the deamidated RLC. Respective mass spec data on native ventricular samples show that ELC can be phosphorylated at 9 residues while RLC at four residues including the known RLC phosphorylation site at Ser15. The authors need to reconcile the mass spectrometry data with Western blotting results showing (+1P) and di-(+2P) phosphorylated ELC and (0P)-RLC. They also should propose which of the nine phosphorylated ELC residues form (+1P) and di-(+2P) ELC.

Thank you for this important point. We first repeated LC-MS from independent tissue samples and could completely verify our previous findings. Since this point is important, we extended our analysis by retrieving peptide spectra from Proteomics DB^{1,2} from myocardial specimen. By this systematic analysis, we could confirm the phospho sites T72, T88, T127, T129 and S179 in completely independent measurements. We also detected T64 and T129, which were published by Arrell *et al.* 2001³ and Cadete *et al.* 2012⁴. Further we identified the ELC phospho sites T113, T147 and T165. Importantly, not all phosphorylation sites will be used simultaneously, we consider stochastic effects of chain phosphorylation and believe that most ELC molecules carry ≤ 3 phosphorylations.

After consulting an additional mass-spectrometry expert (Prof. Jeroen Krijgsveld, EMBL) the data interpretation was confirmed and is highly consistent across different experiments. In theory, there is a slight chance of posttranslational sulfation on some of the identified residues, that yield the same charge

and nearly the same mass (79.9663 vs. 79.9568). Hence, the phosphatase experiments were repeated and are critical for the interpretation. Importantly, successful *in vitro* dephosphorylation was demonstrated by using a phospho-Troponin I antibody (Ser23/24) (Figure 6 F). The highly reproducible experiment clearly underlines at least two active, simultaneous and highly abundant phosphorylations at the sites of ELC and one of RLC. The deamidation events of ELC and RLC could also be verified by our MS, quantitation and functional consequences of such deamidations are, however, not in the focus of our manuscript. The limitation of our quite extensive approach will remain that it is not able to pinpoint the quantitative use of each detected phosphorylation site, which would require generation of phospho-specific antibodies for all nine phosphorylation sites. For comparing DCM and control probands for the ELC phosphorylation, we performed a series of additional LC-MS, showing the distinct centers of highest phosphorylation across ELC (Figure 7 E). Together, we provide extensive new data that we carefully generated following your advice and which we believe clearly underlines our previous assumptions.

The use of mass spec could also be helpful to show the pattern of phosphorylated ELC residues after the samples were treated with phosphatase. Experiments also showed an enhanced ELC phosphorylation in DCM samples compared with controls (Figure 6B), and -as before - the phosphorylated (or charged) ELC variants disappeared on phosphatase treatment. It would be informative to acquire mass spec data on DCM samples \pm phosphatase.

The relative rare event of detecting phospho-peptides after 2D electrophoresis and spot picking results in insufficient quantitation capability of small areas by LC-MS. We, however, visualized the sum of phospho peptides normalized to the total peptide counts detected in spot areas assigned to ventricular ELC (Figure 7 E). Comparing DCM and control samples, we show a significant increase of phospho-peptides in DCM. We also followed your advice and picked the region of ELC in DCM with or without prior treatment with phosphatase. We see more peptides from ELC carrying a phosphorylation (63%) as compared to treated samples (24%). Hence, this confirms our 2D immunoblot experiments that are able to quantify the two most abundant phosphorylation events, underlined by sensitivity analysis and calibration curves (Figure 6 D, E). As also delineated below, this method is suitable for characterizing ELC, which is in a dense region of proteins and which is difficult to interpret using conventional 2D or 1D methods.

Regarding the RLC data, the James Stull group proposed that myosin RLC is constitutively phosphorylated (at Ser15) in the heart at a physiological level of approximately 40% (Chang, Battiprolu et al. 2015). The samples presented as "native" in Figure 5F were incubated with the phosphatase-specific inhibitor, and it is unlikely that all charged RLC variants in Figure 5F represent deamidated RLC and none the (+1P)-RLC. Western blotting data are not consistent with mass spec results that identified phosphorylated Ser15-RLC and other P-residues of RLC in normal healthy ventricular samples but no deamidated Asn14 was found (Figure 5G). This controversy should be somehow resolved. Unlike for the ELC, there are commercially available phospho-specific RLC antibodies, and the lack of reaction to phosphatase treatment shown in Figure 5F should be demonstrated by Western blotting and phospho-specific RLC antibodies. It would also be interesting to see the RLC phosphorylation status in DCM samples \pm phosphatase.

We do agree and we do find the deamidation events in our MS data and present them in the revised Figure 6C. Although RLC is definitely not in the focus of our work, we followed your advice and performed the 2D immunoblot with very fresh human tissue (which is quite limited) and show the results in Figure 6F. As shown, there is a significant reduction of the phospho-spot with a weaker remaining modification. In the same aliquoted samples, we provide the ELC modifications and used phospho-troponin I antibody (Ser23/24). In accordance to Chang *et al.* 2015⁵, we quantified two independent 2D RLC immunoblots showing that the remaining spot after dephosphorylation consists of around $37 \pm 6\%$ of the total RLC protein detected. We also visualized the RLC deamidation events in our 4 DCM patients and 2 control samples analysed by 2D LC-MS (Figure below).

Figure: Normalized amount of RLC deamidated peptides shown for the respective RLC deamidation site. (DCM: n = 4; Ctrl: n = 2). All data are mean \pm SEM.

In summary, the authors are encouraged to provide direct evidence for (+1P)-ELC and (+2P)-ELC and identify the phosphorylated residues. The phosphatase treatment data on charged variants of ELC should be supported by direct evidence showing that 2D SDS-PAGE spots are truly mono- and di-phosphorylated ELC. The phosphatase treatment experiment should include positive controls with the use of other than ELC phosphorylated proteins whose dephosphorylation due to phosphatase treatment could be clearly documented.

See our comments above and revised manuscript.

With the lack of commercially available phospho-specific ELC antibodies, the phosphorylation of ELC may be confirmed using radioisotope phosphate labeling of protein with radioactive ATP. Other methods could include the Pro-Q Diamond phosphoprotein gel stain that selectively stains phosphoproteins in polyacrylamide gels. Although as stated by the authors "1D-phosphogels were not able to separate different ELC forms from human tissue", they might have tried this method already, but it is not stated in the manuscript. There are numerous ProQ/Sypro images of ventricular tissue from healthy and diseased hearts available in the literature showing a positive reaction to ProQ of phosphorylated forms of MyBP-C, Troponin T, and Troponin I, Tropomyosin, and myosin RLC. This reviewer is not aware of any ProQ/Sypro images that were able to detect phosphorylated ELC. Likewise, there are many published images of phosphorylated RLC using Urea/glycerol gels in which the phosphorylated form of RLC migrates faster than the non-phosphorylated RLC. None of such images exist to demonstrate ELC phosphorylation.

RLC is located in a relatively protein-free region of the 2D page, while ELC is not. This was the challenge from the start of our ELC work, which we addressed by using specific ELC antibodies in combination with 2D page. As shown by MS, we find more concurrent proteins in the area of ELC compared to RLC (Supplemental Figure 7). We followed your advice and also present the Pro-Q experiments (Supplemental Figure 8) that underline not only the contribution of phosphorylation to RLC, but also provides evidence of ELC phosphorylation, however, as indicated there is a too big overlap with other proteins and sensitivity is much better in the immunoblot experiments.

On another issue, it was observed that in response to ionophore stimulation and increased [Ca²⁺], the phosphorylation of ELC is significantly enhanced and an additional phosphorylated ELC form (+2P) becomes detectable (Figure 3D). The authors do not discuss any potential mechanism for this observation. Does Ca²⁺ bind to NEK9 activating the kinase in a Ca²⁺ concentration-dependent manner?

Our experiment in Figure 2 D, E underlines that the interaction of ELC and NEK9 increases with increasing intracellular calcium concentration. We could locate the spot of most likely interactions to the kinase domain of NEK9 and believe that calcium improves the binding likelihood as shown for many other protein-protein interactions.^{6,7} By the increased interaction, we postulate the increased activity on ELC phosphorylation.

Reviewer #4:

Although the authors presented a revised manuscript with new information, many major concerns remain. The authors could not meet my concerns about identification of ELC phosphorylation. One main experiment of the authors to proof that ELC is mono- and diphosphorylated (1P, 2P, resp.) is dephosphorylation of ELC and RLC with a phosphatase (Figures 3C, 5E, and 5F). It is clear from Fig. 3C and 5F that loading of ELC after phosphatase treatment is much smaller than ELC loading in the native state. The presence of phosphorylated spots, therefore cannot be detected. This could easily explain the lack of 1P and 2P after phosphatase treatment and has nothing to do with ELC dephosphorylation.

We do appreciate the critique of this referee, however, suggesting that the loading of our protein PAGE was inaccurately is completely incorrect. We repeated the cited experiment (new Figure 7 C) and also provide additional blots with shared protein aliquots and housekeeping protein visualization (Figure 6 F). We also want to point out the changes of phosphorylation in the NEK9 knockdown and overexpression experiments were always normalized to the total ELC levels and represented as ratio to the most basic spot. We do not compare different 2D immunoblots with each other, but only measure within one gel to avoid normalization bias. We now also include LC-MS data showing the changes in ELC phosphorylation in DCM versus controls. As shown in Figure 7 E, the phospho-peptides in diseased and healthy hearts are compared and spectra are available for review by this referee.

Furthermore, I doubt that the phosphatase activity was sufficient for light chain dephosphorylation. This can be deduced from Figure 5C, i.e. native and dephosphorylated RLC. Protein loading in this Figure was comparable between native and phosphatase treated RLC, but no effect of phosphatase treatment

on RLC phosphorylation could be demonstrated in this experiment. This shows that the phosphatase experiment performed by the authors failed in general. The authors left the referee with the question of the nature not only of the ELC but also of the RLC spots.

As pointed out by reviewer #3 there is a deamidation event in RLC that partly or completely (depending on first dimension separation) overlaps with the +1P RLC (see also new Figure 6 F). We hence reperformed this experiment also for RLC from fresh human tissue and complemented this with additional MS data (Figure 6 F and Supplemental Figure 8). As shown, RLC shifted approximately 0.1 pH sensitive to alkaline phosphatase treatment, which is equivalent to a phosphorylation event. For ELC the respective experiment is shown in Figure 7 C.

It was nice to see that the authors could confirm the presence of multiple (9) ELC phosphorylation sites within the ELC sequence (Fig. 5G). But the referee was rather disappointed that the authors did not even try to identify the phosphorylated amino acid sites supposed to be ELC- 1P and 2P.

We thank for recognizing our many efforts to identify and characterize a preciously elusive ELC kinase. We also are first to describe the complex nature of ELC phosphorylation sites and their change in disease and upon extrinsic manipulation. We, however, would need to generate 9 different ELC phospho-antibodies to really map the status of phosphorylations. To address your question, we measured the abundance of ELC peptides carrying a phosphoresidue by LC-MS (Figure 7 E). Since these are digested ELC molecules, not all phosphorylations will/must be present at the same time. There are peaks of phosphorylation at residues T72, T88, T127 and T147. We are convinced that this novel insight together with the identification of NEK9 and its characterization in three genetic model systems is important for the community.

Furthermore, in case there are 9 amino acid sites phosphorylated, one would expect to see 10 ELC protein spots in the 2D-PAGE, far above the 1P and 2P presented in the revized manuscript.

The detection of phospho peptides by LC-MS is very sensitive and does not automatically mean that they reach a certain abundance for detection on 2D page. Our data suggests that DCM patients have 3-4 more abundantly phosphorylated residues. Since they are tryptic digests, it is not said that they are occurring in one ELC molecule at all. See also comment below.

Likewise, the authors showed three more amino acid sites phosphorylated, besides the known S15 of RLC. If this is true, then the RLC should show 8 protein spots in the 2D-PAGE, far more than the two spots shown in Figure 5F. The authors did not give any reasonable explanation of these obvious contradictions.

We do not agree with this interpretation. It is the same issue that some of the modifications might be rarer, but detectable by MS. We do show the calibration curves of our 2D immunoblot (Figure 6 D, E) and obviously there is a limit of detection (sensitivity) for this approach, as is for the Pro-Q-staining and all other detection method. This does, however, not hinder the correct quantification of more abundant phospho-events.

There are some more major concerns. The authors claimed to have identified kinases which phosphorylate ELC. But it is not clear at all which ELC amino acid sites were phosphorylated by these kinases. In addition, closely mapped protein-protein interaction sites between ELC and the kinases should be presented. This are basic information required for publication in a major journal.

We agree. We performed your suggested experiment in two species, zebrafish and humans. As shown in Figure 4 H and I, there is a decreased interaction of piscine ELC and NEK9^{78del} lacking the ATP binding domain located within the protein kinase domain. We went further and generated four different truncation mutants of human NEK9 and mapped the immuno-interaction with human ELC. As shown in the quantification, there is a dip of interaction in the kinase deficient version of NEK9 and also in the version missing the RCC domain. The other two mutations do not change the interaction intensity. We visualized the interaction sites in the three-dimensional structure of NEK9 for illustration purposes (Figure 4 G). Importantly, both domains which we identified have been shown to be interacting with known non-cardiac kinase-targets of NEK9.

With Figure 3D the authors tried to show that NEK9 activity is calcium dependent: ELC phosphorylation under basal condition rose with increasing calcium, an effect which disappeared in case of NEK9 knock down. However, if NEK9 is overexpressed, ELC phosphorylation rose somewhat even without calcium, while additional treatment with calcium was without any effect on ELC phosphorylation. Again, this contradiction is not explained at all.

This might be a typical observation for a saturation effect of kinase-target interactions. In our performed *in vitro* experiments the kinase is present in excess and further stimulation of the kinase activity by modulators of protein-protein interaction (such as calcium) may not be able to further increase the target phosphorylation. We discuss this important point.

1. Samaras, P., *et al.* ProteomicsDB: a multi-omics and multi-organism resource for life science research. *Nucleic Acids Res* **48**, D1153-D1163 (2020).
2. Schmidt, T., *et al.* ProteomicsDB. *Nucleic Acids Res* **46**, D1271-D1281 (2018).
3. Arrell, D.K., Neverova, I., Fraser, H., Marban, E. & Van Eyk, J.E. Proteomic analysis of pharmacologically preconditioned cardiomyocytes reveals novel phosphorylation of myosin light chain 1. *Circ Res* **89**, 480-487 (2001).
4. Cadete, V.J., *et al.* Ischemia/reperfusion-induced myosin light chain 1 phosphorylation increases its degradation by matrix metalloproteinase 2. *FEBS J* **279**, 2444-2454 (2012).
5. Chang, A.N., *et al.* Constitutive phosphorylation of cardiac myosin regulatory light chain *in vivo*. *J Biol Chem* **290**, 10703-10716 (2015).
6. Lee, K.H., Lee, S., Lee, W.Y., Yang, H.W. & Heo, W.D. Visualizing dynamic interaction between calmodulin and calmodulin-related kinases via a monitoring method in live mammalian cells. *Proc Natl Acad Sci U S A* **107**, 3412-3417 (2010).
7. Santamaria-Kisiel, L., Rintala-Dempsey, A.C. & Shaw, G.S. Calcium-dependent and -independent interactions of the S100 protein family. *Biochem J* **396**, 201-214 (2006).

Reviewers' Comments:

Reviewer #1:

Remarks to the Author:

The authors should be commended on the considerable efforts made to produce new experimental data to support their findings and address the concerns of the reviewers.

A common theme among the reviewers reports is the difficulty in reconciling 2-D Western blot data with the evidence from mass spectrometry data. The authors correctly point out that phospho-site usage on any given protein is a distribution (ie from 0-9 for ELC), and that only the most abundant combination will be within the sensitivity of antibody detection (in this case up to 2 sites). The identity of these sites cannot be established by this method, but the spectra showing site utilization (Supp Figure 6) is overall convincing.

I have only minor comments regarding this data - is there a confidence score assigned to these phosphosites in Proteome Discoverer and if so what filters were applied (please amend Methods accordingly). Further, if previously published spectra were retrieved from a public database, would it be possible to align matching peaks eg using mirror plots?

Reviewer #2:

Remarks to the Author:

My previous concern remains on Fig. 4E. It seems impossible that the two heterozygous fish manifest heart failure phenotypes at 72 hr. Did these heterozygous fish later recover from heart failure and survive to adult? If not, you cannot obtain stable heterozygous mutants. Something is wrong here. What are fish number (N) for each bar? Have you determined genotyping? This need to be stated in the text.

Also, in your rebuttal letter: "In addition, at 72 hpf only 8% of the characterized embryos after incrossing of the two different NEK9 mutant lines (Incross nek978del/+ x nek9+/500del) were combined heterozygous mutants (nek978del/500del), indicating your point of a more detrimental effect in combined heterozygosity (Supplemental Figure 4 C)."

Again, what is the total number? 8% also appears unreal, because it is expected that ~25% offsprings are nek978del/500del, which is independent on whether they are embryonic lethal or not. This statement implies that double heterozygous mutants die before 72 hr, which is impossible.

Reviewer #3:

Remarks to the Author:

Remarks to authors:

The revised manuscript by Müller et al. is much improved. The authors provided more LC-MS data from independent tissue samples that verified previous findings. Importantly, a dephosphorylation reaction was performed using additional controls including TnI and a phospho-TnI antibody against Ser23/24. They provided new LC-MS data on DCM vs. ctrl samples and showed centers of highest phosphorylation across ELC confirming enhanced phosphorylation in DCM samples (Fig. 7 E).

Although I am satisfied with the authors' responses, there is still a remaining issue with the data presented in Figure 3. In the previous version of the manuscript, the authors compellingly showed on 2D SDS-PAGE that after the treatment with phosphatase the two charged ELC spots, which I thought were identified with human ELC-specific antibodies, were no longer present, suggesting that the charged spots seen on the gel before the treatment with phosphatase were mono (+1P) and di (+2P) phosphorylated ELC. I then suggested that to validate these data, the membrane in Fig. 3C could be blotted with myc-specific antibodies as the myc-tagged ELC was overexpressed in HEK293 cells. In response, the authors present the same as previously blot claiming that in fact a myc-antibody was used all along. No acknowledgment of the previously mislabeled figure is provided. To clarify this issue and provide a clear picture of ELC phosphorylation in HEK293 cells,

the authors ought to perform a new 2D SDS-PAGE which they will simultaneously blot with rabbit polyclonal antibody against human ELC (Biozol, GeneTex, ZF127578) and monoclonal antibody against myc-tag (Cell signaling, #2040). Both antibodies are listed in the reagents pool. The blot should also have an MW marker and two standard proteins, human ventricular ELC and myc-ELC as controls. By using a human ELC antibody, the ELC phosphorylation pattern will not be confounded by detecting any intrinsic HEK293 ELC proteins.

Minor:

1. Interesting new data are provided on phosphorylated vs. deamidated RLC (Fig. 6F); however, the high degree of deamidated RLC in the samples treated with phosphatase ($37 \pm 6\%$) needs to be further confirmed by using commercially available phosphospecific RLC antibodies.
2. It would be important to know at what Ca^{2+} concentration the NEK9 kinase becomes activated to significantly increase the kinase-target interaction.
3. The newly identified phosphorylation site at T127 in human ventricular ELC sequence has been previously reported by Cadete et al. FEBS 2012. The site was identified by in vitro assay in samples of MLCK phosphorylated human ventricular ELC protein and ex-vivo in rat hearts where equivalent T132 in rat ventricular sequence was identified to be phosphorylated.
4. Please correct the shifted labels in Fig. 4H so they match the presented IB.

Reviewer #4:

The authors corrected their manuscript and provided some new and interesting information. But some major and minor problems still remain.

Major problems:

Many of my questions could not be addressed convincingly by the authors. It is still not clear which amino acids out of the 9 detected phosphoaminoacids of the ELC are phosphorylated to form mono- and diphosphorylated forms. I agree that some of the 9 phosphoaminoacids detected in ELC may be less abundant, less functionally relevant, and may not be detected by 2D. Some, however may be abundant, physiologically relevant, and detectable by 2D while still not analyzed by the authors.

Closely mapped protein-protein interaction sites between ELC and kinase are still missing. The authors presented evidence for ELC-binding sites on NEK9. However, NEK9 binding sites of ELC were not demonstrated.

Minor problems

Abstract: 1st sentence is unspecific and should be deleted.

Figure 2B shows different results than that described in the Results section of the manuscript. „NEK9 is highly expressed in human heart tissue on then mRNA (Figure 2 A) and protein level (Figure 2 B) as well as in human skeletal muscle“. In fact, there is no human skeletal muscle lane for NEK9.

Point by point response

Reviewer #1

The authors should be commended on the considerable efforts made to produce new experimental data to support their findings and address the concerns of the reviewers.

Thank you very much for recognizing our effort and for your positive perception of our revised manuscript.

*A common theme among the reviewer's reports is the difficulty in reconciling 2-D Western blot data with the evidence from mass spectrometry data. The authors correctly point out that phospho-site usage on any given protein is a distribution (i.e. from 0-9 for ELC), and that only the **most abundant combination** will be within the sensitivity of antibody detection (in this case **up to 2 sites**). The identity of these sites cannot be established by this method, but the spectra showing site utilization (Supp Figure 6) is overall convincing.*

Thank you again for supporting our data interpretation.

I have only minor comments regarding this data - is there a confidence score assigned to these phosphosites in Proteome Discoverer and if so what filters were applied (please amend Methods accordingly). Further, if previously published spectra were retrieved from a public database, would it be possible to align matching peaks eg using mirror plots?

Target FDR for PSM and peptides were set to 0.01 and results were filtered for high confident peptides. The information was added to the Methods section. For the already published data that we used, we refer to the original data depositions in our manuscript.

Reviewer #2:

My previous concern remains on Fig. 4E. It seems impossible that the two heterozygous fish manifest heart failure phenotypes at 72 hr. Did these heterozygous fish later recover from heart failure and survive to adult? If not, you cannot obtain stable heterozygous mutants. Something is wrong here.

We appreciate the reviewer's point. We injected Cas9 and gRNA targeting the piscine *nek9* gene. The injected fish were raised to adulthood, outcrossed with wild type fish and the F1 generation were genotyped by fin biopsy (Supplemental Fig. 5 B). We identified two different heterozygous transgenic *nek9* lines with different deletion events, either harboring a 78 bp or 500 bp deletion of the zebrafish *nek9* gene (Supplemental Fig. 5 C). First, we deeply analysed the functional effects of the mutation by incrossing the two CRISPR/Cas9 lines (*nek9^{+/500del}x nek9^{78del/+}*). Importantly, the entire clutch of zebrafish embryos (offspring from one mating) was phenotypically characterized at 72 hpf. All embryos were checked for chamber morphology, blood flow, beating abnormalities etc. and hearts were recorded to determine fractional shortening. Afterwards, the embryos were collected in a well plate with defined numbering/position for genotyping. The identified genotype was assigned to the previously documented phenotype for each fish and the penetrance was analyzed. As shown in Fig. 4 E, the alleles are different with regard to their penetrance of heart failure, with *nek9^{500del}* resulting in higher rates of heart failure compared to the heterozygous *nek9^{78del}* fish. This is a well-known phenomenon in humans.

In addition, we were able to raise a sufficient number of heterozygous adult *nek9* CRISPR/Cas9 fish to carry them (*nek9^{78del/+}* x *nek9^{78del/+}*, *nek9^{+/500del}* x *nek9^{+/500del}*). As shown in Supplemental Fig. 6 A, in an early state at 24 hpf the genotype ratio is closer to the expected Mendelian ratio with 34 % wild type; 24 % heterozygous and 15 % homozygous zebrafish (27 % of fish were dead). After 72 hpf, no homozygous fish embryos had survived, confirming a lethal effect of both homozygous mutant alleles. In contrast, some combined heterozygous zebrafish embryos survive until 72 hpf (24 hpf: 12%; 72 hpf: 8%) (Supplemental Fig. 6 A, Fig 5 D-F). Nevertheless, we could not identify adult combined heterozygous zebrafish. Phenotypic analysis after 72 hpf revealed significantly decreased fractional shortening with the same severity in *nek9^{78del/+}* and *nek9^{+/500del}*, while the penetrance was higher in the *nek9^{+/500del}* mutant embryos (Supplemental Fig. 6 B, C). Taking together, analysis of the two mutated alleles confirmed the data of the outcross experiments showing incomplete penetrance for both heterozygous alleles, which enables enough embryos to survive to adulthood. The severity of the heart failure is comparable in both lines and homozygous carriers are embryonic lethal. We followed your valuable advice and clarified these important points in the paper.

What are fish number (N) for each bar? Have you determined genotyping? This need to be stated in the text.

In all figures labeled with a genotype (*nek9^{+/+}*, *nek9^{78del/+}*, *nek9^{78del/78del}*, *nek9^{+/500del}*, *nek9^{500del/500del}*, *nek9^{78del/500del}*) genotyping was performed by using specific primer pairs as stated in the Methods section. We added the sample size to the figure legends and provided detailed information on statistical calculations. Importantly, to avoid technical bias, we analyzed the entire clutch of zebrafish embryos phenotypically and afterwards genotyped each fish embryo in a blinded manner (which is quite a lot of work, when you see all the n). Association of genotype with the corresponding phenotype was done at the end, so the whole process was blinded for the investigator!

*Also, in your rebuttal letter: "In addition, at 72 hpf only 8% of the characterized embryos after incrossing of the two different NEK9 mutant lines (Incross *nek9^{78del/+}* x *nek9^{+/500del}*) were combined heterozygous mutants (*nek9^{78del/500del}*), indicating your point of a more detrimental effect in combined heterozygosity (Supplemental Figure 4 C)." Again, what is the total number? 8% also appears unreal, because it is expected that ~25% offsprings are *nek9^{78del/500del}*, which is independent on whether they are embryonic lethal or not. This statement implies that double heterozygous mutants die before 72 hr, which is impossible.*

We added the total number of analysed embryos to the figures. As presented in Supplemental Fig. 6 A, we found 10 out of 82 embryos at 24 hpf (12%) and 19 out of 247 embryos to be compound heterozygous at 72 hpf (8%). As also shown for the incross experiments, the expected 25% homozygous mutation carriers or combined heterozygous embryos are not observed. Genotype analysis clearly shows no homozygous embryos at 72 hpf, while some could be detected at an early stage of 24 hpf, indicating embryonic lethality. Combined heterozygous zebrafish were found at 72 hpf, but could not be raised to adulthood. As a side note, homozygous NEK9 knockout mice are embryonic lethal (MGI: 2387995).

Reviewer #3

The revised manuscript by Müller et al. is much improved. The authors provided more LC-MS data from independent tissue samples that verified previous findings. Importantly, a dephosphorylation reaction was performed using additional controls including Tnl and a phospho-Tnl antibody against Ser23/24. They provided new LC-MS data on DCM vs. ctrl samples and showed centers of highest phosphorylation across ELC confirming enhanced phosphorylation in DCM samples (Fig. 7 E).

Thank you for your positive and constructive comments, which were instrumental to further improve our manuscript.

Although I am satisfied with the authors' responses, there is still a remaining issue with the data presented in Figure 3. In the previous version of the manuscript, the authors compellingly showed on 2D SDS-PAGE that after the treatment with phosphatase the two charged ELC spots, which I thought were identified with human ELC-specific antibodies, were no longer present, suggesting that the charged spots seen on the gel before the treatment with phosphatase were mono (+1P) and di (+2P) phosphorylated ELC. I then suggested that to validate these data, the membrane in Fig. 3C could be blotted with myc-specific antibodies as the myc-tagged ELC was overexpressed in HEK293 cells. In response, the authors present the same as previously blot claiming that in fact a myc-antibody was used all along. No acknowledgment of the previously mislabeled figure is provided. To clarify this issue and provide a clear picture of ELC phosphorylation in HEK293 cells, the authors ought to perform a new 2D SDS-PAGE which they will simultaneously blot with rabbit polyclonal antibody against human ELC (Biozol, GeneTex, ZF127578) and monoclonal antibody against myc-tag (Cell signaling, #2040). Both antibodies are listed in the reagents pool. The blot should also have an MW marker and two standard proteins, human ventricular ELC and myc-ELC as controls. By using a human ELC antibody, the ELC phosphorylation pattern will not be confounded by detecting any intrinsic HEK293 ELC proteins.

To provide further evidence for interaction of NEK9 and ELC and to show the impact on the ELC phosphorylation status, we used an *in vitro* system, which excludes intrinsic interference. HEK293 are suitable, as the cell line is originally derived from human embryonic kidney cells and fails to express cardiac ELC¹ (see also Fig. 1 E, 2 D, 4 H, Supplemental Fig. 2 and 7). Exogenous myc-tagged protein further improves specificity by additionally using antibodies against the myc-tag. Importantly, the myc-tagged human ELC completely resembles the ELC phosphorylation pattern in human myocardium, as shown by dephosphorylation experiments before (Fig. 3 C). In all experiments myc-tagged human ELC was used (Fig. 1 E, 2 D, 3 C-E, Supplemental Fig. 2 and 7). In Fig. 4 H, we performed protein-protein interaction experiments and used untagged piscine ELC and myc-tagged piscine NEK9 protein of the different mutated zebrafish lines. We updated the Methods section for accuracy. In addition, we followed your advice and performed new 2D IBs to validate our *in vitro* system. After overexpression of myc-tagged human ELC in HEK293 cells and performing Ca²⁺-influx by ionophore stimulation, we run 2D IB and stained the membrane successively with myc-tag antibody (Cell signaling #2040), β-actin antibody (Cell signaling #3700) and human ELC antibody (Biozol, ZF127578) (Supplemental Fig. 2). Membranes were reblotted by using a standard protocol described in the Methods section. Proteins of untransfected, Ca²⁺ treated cells were run as a control in the same experiment. Staining with the myc-tag antibody or

the human ELC antibody clearly detected overexpressed ELC and resulted in the same ELC phosphorylation pattern showing one basic ELC protein form and two abundant phospho ELC forms shifted towards an acidic pH. Human ELC antibodies as well as myc specific staining are highly specific. Untransfected HEK293 cells did not show any ELC nor myc-tag signal, while equal amounts of the standard protein β -actin were detected in the transfected compared to the untransfected sample. Of note, the standard proteins needed to be in the detection range between pH 4.5 and 5.5 (β -actin: MW 42kDa, pI 4.29). All 2D IBs in the presented manuscript were run with a standardized protein ladder (PageRuler™ prestained, ThermoFisher #26617). The new blots presented in Supplemental Fig. 2 show an additional marker on the right site (Precision Plus Protein™ WesternC™ Blotting Standards, Biorad #1610385). We believe the data of the *in vitro* experiment are strong, since we can analyze the ELC phosphorylation pattern without intrinsic protein influence, while completely mimicking ELC phosphorylation in human myocardium and resembling the other experimental, mechanistic data. We apologize for any confusion that might have been caused by naming inconsistencies between methods and figures.

Minor:

1. Interesting new data are provided on phosphorylated vs. deamidated RLC (Fig. 6F); however, the high degree of deamidated RLC in the samples treated with phosphatase ($37 \pm 6\%$) needs to be further confirmed by using commercially available phosphospecific RLC antibodies.

The weaker remaining modification after treatment with phosphatase in Fig. 6 F might be a deamidation, as these modifications were most likely detected by 2D LC-MS. Nevertheless, it cannot be ruled out, that other modifications or even protein isoforms correspond to the spot. Of note, the RLC dephosphorylation experiments underline the limitation of the 2D phosphorylation assay as insufficient to quantify RLC phosphorylation status in our patient cohort. Of note, we focus on NEK9-ELC, not RLC.

2. It would be important to know at what Ca^{2+} concentration the NEK9 kinase becomes activated to significantly increase the kinase-target interaction.

In our *in vitro* experiments using HEK293 cells, the cells were treated with increasing ionophore concentrations (1 to 5 μ mol/L) in a saturated environment of 4 mmol/L $CaCl_2$. This is in concordance with the literature mimicking Ca^{2+} influx.²⁻⁵ Kinase activity can be regulated indirectly by Ca^{2+} dynamics or directly by Ca^{2+} binding. For both mechanisms, the exact Ca^{2+} concentration might be less important than the ion flux resulting in a change of the intracellular Ca^{2+} concentration. In contrast to the calcium-calmodulin dependent kinases (CamK), which can bind the maximum of four Ca^{2+} ions⁶, we are not expecting direct binding of Ca^{2+} to NEK9. ExPASy Prosite protein domain search did not predict a known Ca^{2+} binding domain.⁷ Hence, we believe that the interaction enhancement is the mode of action in case of NEK9-ELC. We added this discussion to the manuscript.

3. The newly identified phosphorylation site at T127 in human ventricular ELC sequence has been previously reported by Cadete et al. FEBS 2012. The site was identified by in vitro assay in samples of MLCK phosphorylated human ventricular ELC protein and ex-vivo in rat hearts where equivalent T132 in rat ventricular sequence was identified to be phosphorylated.

Thank you for your advice. We added this information to the manuscript.

4. Please correct the shifted labels in Fig. 4H so they match the presented IB.

Thanks for your note. We corrected the visualization.

Reviewer #4

The authors corrected their manuscript and provided some new and interesting information. But some major and minor problems still remain.

We thank the reviewer for the comments on our manuscript.

Major problems:

Many of my questions could not be addressed convincingly by the authors. It is still not clear which amino acids out of the 9 detected phosphoaminoacids of the ELC are phosphorylated to form mono- and diphosphorylated forms.

In our manuscript, we detected increased ELC phosphorylation in LV tissue of DCM patients compared to controls (Fig. 7 B). We performed 2D IBs in a deeply phenotyped patient cohort of 11 DCM patients and 9 controls (including 3 healthy controls) (Fig. 7 B, C). Phosphorylated ELC protein species are shifted towards an acidic IP and could be separated from non-phosphorylated protein species (Basal) on 2D Gel. We and others could identify 2 shifted ELC protein spots corresponding to phosphorylated ELC species and labeled them with (+1P and +2P). Validation of our assay was done by dephosphorylation experiments, which clearly show one remaining non-phosphorylated ELC spot (Fig. 3 C and Fig. 7 C). By running calibration curves for the ELC antibody (Fig 6 D, E) a linear detection range was identified, which enables quantification of the ELC phosphorylation. To go even further, we picked the ELC protein spots and analysed them by LC-MS. We detected 9 amino acid residues of ELC as phosphorylated (Fig. 6 C). Importantly, we are not expecting 9 ELC phospho-spots on 2D IB, as not all phosphorylation sites are used simultaneously. We considered stochastic effects of chain phosphorylation and believe that most ELC molecules carry ≤ 3 phosphorylations. However, a limitation of the LC-MS method is the sensitivity to detect phospho-peptides after silver staining and tryptic digestion. Pinpointing the exact phospho peptides corresponding to the phospho ELC spots on the 2D gel would need 9 phosphospecific antibodies. Nevertheless, in a validation cohort of 4 DCM patients and 2 healthy control samples we could confirm a significant increase of ELC phospho-peptides in DCM patients in LC-MS (Fig. 7 E) and by visualization of the detected phospho-peptides along the ELC protein we identified four amino acid residues of high abundance (T72, T88, T127 and T147). Phospho peptides at position T72 and T147 were significantly increased in DCM patients compared to controls (Fig. 7 E). We are convinced that the deep analysis of the ELC phosphorylation in human DCM patients is of high interest for the community and could be the subject of a future study.

I agree that some of the 9 phosphoaminoacids detected in ELC may be less abundant, less functionally relevant, and may not be detected by 2D. Some, however may be abundant, physiologically relevant, and detectable by 2D while still not analyzed by the authors.

We appreciate this point, however, the functional characterization of all 9 detected phosphorylation sites is out of the focus of our current manuscript. Furthermore, we do not think, that functional characterization of the identified ELC phosphorylation site independently, for example by genetic manipulation, would lead to meaningful data. The posttranslational modification of sarcomeric proteins, especially phosphorylation, is highly dynamic and it is known, that there is an interplay between different modifications affecting force generation.⁸ Further studies might focus on the overall phosphorylation status of the sarcomere and concentrate on manipulation of upstream targets like kinases or phosphatases responsible for posttranslational phosphorylation.

Closely mapped protein-protein interaction sites between ELC and kinase are still missing. The authors presented evidence for ELC-binding sites on NEK9. However, NEK9 binding sites of ELC were not demonstrated.

We performed protein-protein interaction experiments by truncating the NEK9 kinase (980 amino acids) instead of the small ELC protein (195 amino acids) as NEK9 is the major focus of our manuscript. Our data show, that heterozygous zebrafish embryos *nek9*^{78del/+} lacking the ATP binding domain develop heart failure reflected by impaired fractional shortening (Fig. 4 D-F). By using piscine NEK9^{78del} protein we observed a decrease of ELC/NEK^{78del} interaction (Fig. 4 H, I). The lack of the ELC/NEK9 interaction could be confirmed for human protein *in vitro*. Human NEK9 protein lacking the kinase domain (truncated amino acid position 52-308) shows a dip of interaction with human ELC (Fig. 4 J). In addition, we analyzed three different truncation mutants of human NEK9 (truncated amino acid position: a: 172-184; b: 727-980; c: 309-726) and mapped the interaction with human ELC.

Minor problems

Abstract: 1st sentence is unspecific and should be deleted.

Thank you, we deleted this sentence.

Figure 2B shows different results than that described in the Results section of the manuscript. „NEK9 is highly expressed in human heart tissue on mRNA (Figure 2 A) and protein level (Figure 2 B) as well as in human skeletal muscle.“ In fact, there is no human skeletal muscle lane for NEK9.

We corrected the sentence. We show NEK9 expression in human skeletal muscle on mRNA level (Fig. 2 A). NEK9 was highly expressed in human heart tissue on mRNA (Fig. 2 A) and protein level (Fig 2 B).

1. Uhlen, M., *et al.* A human protein atlas for normal and cancer tissues based on antibody proteomics. *Mol Cell Proteomics* **4**, 1920-1932 (2005).
2. Hu, Y., *et al.* Uncovering the arrhythmogenic potential of TRPM4 activation in atrial-derived HL-1 cells using novel recording and numerical approaches. *Cardiovasc Res* **113**, 1243-1255 (2017).
3. Mori, M.X., Imai, Y., Itsuki, K. & Inoue, R. Quantitative measurement of Ca(2+)-dependent calmodulin-target binding by Fura-2 and CFP and YFP FRET imaging in living cells. *Biochemistry* **50**, 4685-4696 (2011).
4. Papanayotou, C., *et al.* Calfacilitin is a calcium channel modulator essential for initiation of neural plate development. *Nat Commun* **4**, 1837 (2013).
5. Iacobucci, G.J. & Popescu, G.K. Ca(2+)-Dependent Inactivation of GluN2A and GluN2B NMDA Receptors Occurs by a Common Kinetic Mechanism. *Biophys J* **118**, 798-812 (2020).
6. Swulius, M.T. & Waxham, M.N. Ca(2+)/calmodulin-dependent protein kinases. *Cell Mol Life Sci* **65**, 2637-2657 (2008).
7. Sigrist, C.J., *et al.* New and continuing developments at PROSITE. *Nucleic Acids Res* **41**, D344-347 (2013).
8. Budde, H., *et al.* The Interplay between S-Glutathionylation and Phosphorylation of Cardiac Troponin I and Myosin Binding Protein C in End-Stage Human Failing Hearts. *Antioxidants (Basel)* **10**(2021).

Reviewers' Comments:

Reviewer #2:

Remarks to the Author:

The authors did a great job and had carefully addressed all of my concerns. They provided fish number for most figures, and added more details on how they carried their experiments and statistical analysis.

However, my concern remains on their data showing that the distribution of the zebrafish mutant offspring differs significantly from Mendelian ratio, as exemplified by Supplemental Figure 6A. I cannot agree with their interpretation of embryonic lethality, i.e. the dead embryos just disappear, such as before 72 hpf. In this reviewer's mind, every single embryo is tractable by a genotyping method, even after the embryo dies at a certain stage of embryogenesis. For most cardiac mutants, the body of homozygous mutant embryos are typically intact after 5 dpf (thus can be used for genotyping), even their hearts manifest very severe heart failure phenotypes.

The most likely reason is their genotyping method, which might not be highly reliable. It is possible that the heteroduplex motility assay failed to reliably distinguish homozygous mutants from heterozygous mutants or WT control in their hands.

It is recommended for the authors to confirm their related zebrafish data by using a more reliable genotyping method. I hope the new data will be consistent with the Mendelian ratio, which would convince the authors to update their concept on embryonic lethality.

Reviewer #3:

Remarks to the Author:

My remaining concerns were satisfactorily addressed in the revised version of the manuscript by Müller et al. The authors provided new data in Supplemental Figure 2 that validated phosphorylation of human myosin myc-ELC overexpressed in HEK cells with myc-tag antibody (Cell signaling #2040) applied concurrently with human ELC specific antibody (Biozol ZF127578). The authors should be commended on their new and exciting results on NEK9-mediated phosphorylation of myosin ELC and its importance for cardiac contractility in health and disease that open new avenues for further research on ELC phosphorylation and set off the stage for novel therapeutic approaches to heart failure.

Reviewer #2

The authors did a great job and had carefully addressed all of my concerns. They provided fish number for most figures, and added more details on how they carried their experiments and statistical analysis.

Thank you very much for the positive perception of our revised manuscript.

However, my concern remains on their data showing that the distribution of the zebrafish mutant offspring differs significantly from Mendelian ratio, as exemplified by Supplemental Figure 6A. I cannot agree with their interpretation of embryonic lethality, i.e. the dead embryos just disappear, such as before 72 hpf. In this reviewer's mind, every single embryo is tractable by a genotyping method, even after the embryo dies at a certain stage of embryogenesis. For most cardiac mutants, the body of homozygous

mutant embryos are typically intact after 5 dpf (thus can be used for genotyping), even their hearts manifest very severe heart failure phenotypes.

We agree that in case of the *nek9* deletion mutants, the expected Mendelian ratio is not met. At 24 hpf we only find 15% homozygous mutant embryos instead of 25%, while 27% of the embryos are degraded at that stage (Supplemental Figure 6 A). However, we do not agree with the interpretation of the referee. For genotyping the whole embryo needs to be used for DNA extraction, so individual longitudinal genotyping is only possible in adults by fin-clipping. A typical deterioration of NEK9 CRISPR/Cas9 mutant embryos is shown below. DNA isolation is not successful in these degraded embryos. Genotyping of these embryos would be only possible before slight signs of degradation occur, which would need continuous real-time observation of hundreds of embryos for around 48 hours. Nevertheless, the deep analysis of different developmental stages and different mating schemes confirm a lethal effect of both homozygous mutant alleles. As shown in Supplemental Figure 6 A, no homozygous fish embryos had survived until 72 hpf. The data shown can be fully explained by the lethal nature of the *nek9* mutation during early embryogenesis, potentially due to expression of NEK9 in non-cardiac tissue and essential functional role in other investigated tissues by controlling e.g. the cell cycle. As such, a global knockout of *nek9* in mice was shown to be 100% embryonic lethal (MGI: 2387995). The concept of embryonic lethality is supported by several studies, such as the publication of Driever *et al.*, who identified 2383 mutations in a zebrafish screen, from which 494 resulted in early embryonic death and degradation (21% of all lines) beginning at 24 hpf.¹

*Figure: Early embryonic lethality observed in *nek9*^{500del} after in-crossing around 18-20 somite stage. By contrast to cardiac-specific genes, *nek9* plays a broader role during embryonic development and hence complete lack results in degradation. In combined heterozygous state the lethality is lower.*

In contrast to the NEK9 mutant line showing early degradation, disruption of cardiac restricted genes usually does not result in degradation, since singular loss of cardiac function can be compensated up to day 5 by oxygen diffusion via the surrounding medium. The corresponding author of the revised manuscript co-published several cardiac specific and non-specific lines, including *lazy susan*², *flatline*³ or *bungee*⁴, and genes, like *nexn*⁵ or *pinch1*⁶ and *pinch2*⁶. We included a paragraph in the discussion section of the revised manuscript to explain the observed embryonic lethality.

The most likely reason is their genotyping method, which might not be highly reliable. It is possible that the heteroduplex motility assay failed to reliably distinguish homozygous mutants from heterozygous mutants or WT control in their hands. It is recommended for the authors to confirm their related zebrafish data by using a more reliable genotyping method. I hope the new data will be consistent with the Mendelian ratio, which would convince the authors to update their concept on embryonic lethality.

As shown in Supplemental Figure 5 A and B, the heteroduplex assay was only used for screening the F0 generation, which is a standard approach in unknown deletion events. The genotyping method for all further genotype-phenotype association studies was based on two fully informative PCR markers specifically designed for the two deletion mutations. Primer pair F1-R1 flanks the 78 bp deletion and primer pair F2-R2 flanks the 500 bp deletion (Supplemental Figure 5 B). For DNA isolation the whole embryo was used to reach a sufficient amount of DNA for genotyping. All alleles (wild-type, *nek9*^{78del} and *nek9*^{500del}) are clearly identifiable and there is no uncertainty regarding any possible genotype resulting from the crossing of these alleles. In the revised manuscript, we have more prominently highlighted the use of these markers in comparison to heteroduplex assays for the first identifying CRISPR-edits at the *nek9* locus.

Reviewer #3

My remaining concerns were satisfactorily addressed in the revised version of the manuscript by Müller et al. The authors provided new data in Supplemental Figure 2 that validated phosphorylation of human myosin myc-ELC overexpressed in HEK cells with myc-tag antibody (Cell signaling #2040) applied concurrently with human ELC specific antibody (Biozol ZF127578). The authors should be commended on their new and exciting results on NEK9-mediated phosphorylation of myosin ELC and its importance for cardiac contractility in health and disease that open new avenues for further research on ELC phosphorylation and set off the stage for novel therapeutic approaches to heart failure.

Danuta Szczesna-Cordary

Dear Prof. Danuta Szczesna-Cordary, thank you for your valuable contribution, which was instrumental to improve our manuscript.

References

1. Driever W, Solnica-Krezel L, Schier AF, Neuhauss SC, Malicki J, Stemple DL, Stainier DY, Zwartkruis F, Abdelilah S, Rangini Z, Belak J and Boggs C. A genetic screen for mutations affecting embryogenesis in zebrafish. *Development*. 1996;123:37-46.
2. Meder B, Laufer C, Hassel D, Just S, Marquart S, Vogel B, Hess A, Fishman MC, Katus HA and Rottbauer W. A single serine in the carboxyl terminus of cardiac essential myosin light chain-1 controls cardiomyocyte contractility in vivo. *Circ Res*. 2009;104:650-9.
3. Just S, Meder B, Berger IM, Etard C, Trano N, Patzel E, Hassel D, Marquart S, Dahme T, Vogel B, Fishman MC, Katus HA, Strahle U and Rottbauer W. The myosin-interacting protein SMYD1 is essential for sarcomere organization. *J Cell Sci*. 2011;124:3127-36.
4. Just S, Berger IM, Meder B, Backs J, Keller A, Marquart S, Frese K, Patzel E, Rauch GJ, Tübingen Screen C, Katus HA and Rottbauer W. Protein kinase D2 controls cardiac valve formation in zebrafish by regulating histone deacetylase 5 activity. *Circulation*. 2011;124:324-34.
5. Hassel D, Dahme T, Erdmann J, Meder B, Hüge A, Stoll M, Just S, Hess A, Ehlermann P, Weichenhan D, Grimmmler M, Liptau H, Hetzer R, Regitz-Zagrosek V, Fischer C, Nurnberg P, Schunkert H, Katus HA and Rottbauer W. Nexilin mutations destabilize cardiac Z-disks and lead to dilated cardiomyopathy. *Nat Med*. 2009;15:1281-8.
6. Meder B, Huttner IG, Sedaghat-Hamedani F, Just S, Dahme T, Frese KS, Vogel B, Kohler D, Kloos W, Rudloff J, Marquart S, Katus HA and Rottbauer W. PINCH proteins regulate cardiac contractility by modulating integrin-linked kinase-protein kinase B signaling. *Mol Cell Biol*. 2011;31:3424-35.